# PROOFBRIDGE: AUTO-FORMALIZATION OF NATURAL LANGUAGE PROOFS IN LEAN VIA JOINT EMBEDDINGS

**Prithwish Jana**[*1], **Kaan Kale**[1], **Ahmet Ege Tanriverdi**[2]
**Cruise Song**[1], **Sriram Vishwanath**[1], **Vijay Ganesh**[*1]

[1]Georgia Institute of Technology, USA    [2]Bogazici University, Türkiye

## ABSTRACT

Translating human-written mathematical theorems and proofs from natural language (NL) into formal languages (FLs) like Lean 4 has long been a significant challenge for AI. Most state-of-the-art methods either focus on theorem-only NL-to-FL auto-formalization or on FL proof synthesis from FL theorems. In practice, auto-formalization of both theorem and proof still requires human intervention, as seen in AlphaProof's silver-medal performance at the 2024 IMO, where problem statements were manually translated before automated proof synthesis.

We present PROOFBRIDGE, a unified framework for automatically translating entire NL theorems and proofs into Lean 4. At its core is a joint embedding model that aligns NL and FL (NL-FL) theorem+proof pairs in a shared semantic space, enabling cross-modal retrieval of semantically relevant FL examples to guide translation. PROOFBRIDGE integrates retrieval-augmented fine-tuning with iterative proof repair, leveraging Lean's type checker and semantic equivalence feedback to ensure both syntactic correctness and semantic fidelity. Experiments show substantial improvements in proof auto-formalization over strong baselines (including GPT-5, Gemini-2.5, Kimina-Prover, DeepSeek-Prover), with our retrieval-augmented approach yielding significant gains in semantic correctness (SC, via proving bi-directional equivalence) and type correctness (TC, via type-checking theorem+proof) across pass@k metrics on MINIF2F-TEST-PF, a dataset we curated. In particular, PROOFBRIDGE improves cross-modal retrieval quality by up to $3.28\times$ Recall@1 over all-MiniLM-L6-v2, and achieves +31.14% SC and +1.64% TC (pass@32) compared to the baseline Kimina-Prover-RL-1.7B.

## 1 INTRODUCTION

In mathematics, ensuring the correctness of proofs is a crucial yet inherently difficult task. Traditionally, mathematicians rely on the peer-review process for *proof verification*, yet as proofs grow increasingly complex, even careful human scrutiny can overlook subtle errors. For instance, in 1989, Kapranov and Voevodsky published a proof connecting $\infty$-groupoids and homotopy types, which was later disproven by Carlos Simpson in 1998; more recently, while formalizing his 2023 paper (Tao, 2023) on the Maclaurin-type inequality, Terence Tao discovered a non-trivial bug. To mitigate challenges of verifying complex proofs, proof assistants and formal mathematical languages like Isabelle (Nipkow et al., 2002), HOL Light (Harrison, 2009), Metamath (Megill & Wheeler, 2019), Lean 4 (Moura & Ullrich, 2021), Peano (Poesia & Goodman, 2023), and Rocq (Sozeau et al., 2025) have been developed, offering a way to create *computer-verifiable formal proofs*. Such formal language (FL) proofs, defined by strict syntax and symbolic logic, enable reliable automated verification guarantees that resolve the inherent ambiguity of natural language (NL) proofs. However, constructing FL proofs is time-intensive and demands both deep mathematical expertise and detailed knowledge of the language and its libraries. Thus, the process remains challenging even for experienced mathematicians, limiting the wider adoption of formal proof assistants and FL proofs.

To simplify the task of writing proofs in FL, two key research directions have emerged: *auto-formalization* and *automated formal proof synthesis (AFPS)*. Auto-formalization targets NL-to-FL translation, but most prior works (Wang et al., 2025; Wu et al., 2026; Jiang et al., 2024; Gao et al.,

[*]Co-corresponding authors. Emails: {`pjana7`, `vganesh`}`@gatech.edu`

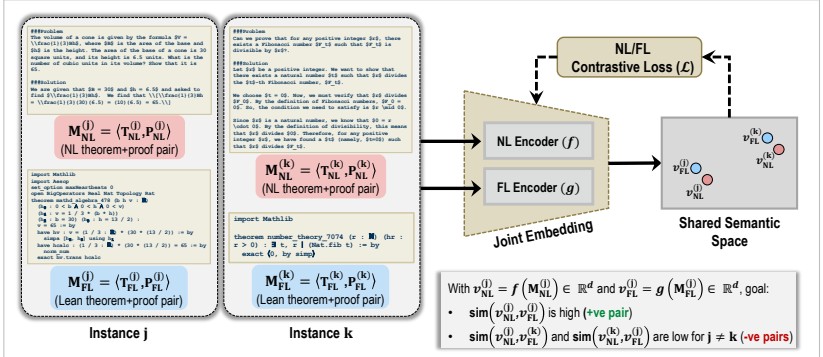

(a) Joint embedding of NL and FL (Lean) theorems and proofs into shared semantic space

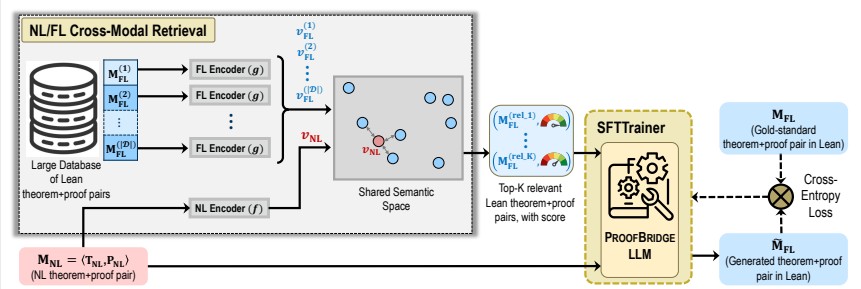

(b) Retrieval-augmented Supervised Fine-Tuning (SFT) of PROOFBRIDGE with NL/FL cross-modal retrieval

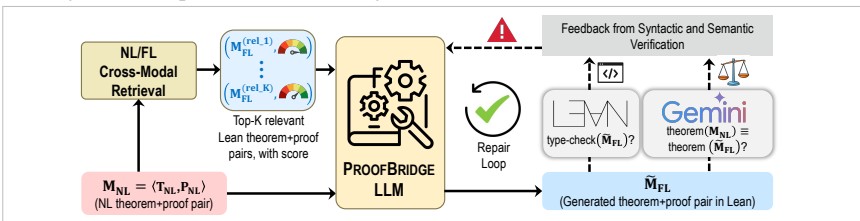

(c) Inference phase of retrieval-augmented proof auto-formalization with iterative repair

Figure 1: **Pipeline of PROOFBRIDGE for proof auto-formalization.** We first train a joint embedding model for NL and FL via contrastive learning, enabling cross-modal retrieval of semantically related FL theorem+proof pairs for a given NL input. An LLM is then fine-tuned on NL-to-Lean translations, conditioned on retrieved proofs and relevance scores. At inference, the system retrieves relevant Lean proofs to guide FL theorem+proof generation and applies an iterative repair loop.

2025) focus only on formalizing theorems (statements), not proofs. In contrast, *AFPS* (Ren et al., 2025; Wang et al., 2025) aims to generate FL proofs given an FL theorem. Proof auto-formalization is relatively less explored, with Draft-Sketch-Prove (Jiang et al., 2023) for Isabelle and FormL4 (Lu et al., 2024) for Lean serving as notable examples. In practice, formalizing an entire NL proof requires first performing *theorem-only auto-formalization* to translate the NL theorem into FL, followed by *AFPS* to generate the FL proof. AlphaProof (Deepmind, 2024), which achieved silver-medal standard in the 2024 International Mathematical Olympiad, followed this two-step process: problems were manually formalized before automated proof synthesis. Thus, in practice, pipelines still require manual theorem formalization prior to proof synthesis, despite advances in theorem-only auto-formalization and AFPS tools. This illustrates the broader challenge that current systems often rely on human intervention to ensure semantic correctness of proof auto-formalization.

Contemporary LLMs face several challenges that limit their effectiveness for proof auto-formalization in Lean 4. **First**, large-scale datasets pairing NL theorems with Lean 4 proofs are scarce. Most existing resources (Goedel-Pset-v1 (Lin et al., 2025), Herald-statements (Gao et al., 2025), Lean Workbook (Ying et al., 2024), MMA (Jiang et al., 2024)) cover only theorems, while those with proofs (Herald-proofs, Lean Workbook proofs (Lin et al., 2025), and FormL4 (Lu et al., 2024)) are much smaller and do not align with the popular miniF2F (Zheng et al., 2022) benchmark in the same Lean 4 version. Lean versions are not backward compatible, so cross-version

evaluation often fails. **Second**, general-purpose foundation models often fail to satisfy strict syntactic and semantic constraints of specialized FLs (Jha et al., 2025; DeLorenzo et al., 2025; Jana, 2024; Ugare et al., 2024), such as Lean, and are computationally expensive (Dasgupta et al., 2025), motivating smaller specialized models. However, most fine-tuned LLMs for Lean target either theorem auto-formalization or proof synthesis. Proof auto-formalization is harder, as it requires both translating the NL theorem and constructing the corresponding FL proof. **Third**, Lean 4 has an effectively *infinite action space* (Poesia & Goodman, 2023), where proofs use complex *tactics* that reuse prior theorems or introduce new variables. Prior work generates FL directly from NL while ignoring semantic relations such as shared tactics and DAG dependencies, often leading LLMs to produce hallucinated or invalid proofs. **Fourth**, automated evaluation is a major bottleneck. Lean's type-checker verifies the FL proof but cannot ensure semantic equivalence. Existing methods often type-check only the theorem (leaving proofs incomplete using placeholders like `sorry`) or rely on proxies such as BLEU, which are unreliable (Lu et al., 2024; Ying et al., 2024; Wu et al., 2022).

**Key Insight.** In this paper, we address the task of *proof auto-formalization*, focusing on Lean 4 as the FL, via a combination of a joint embedding model, an LLM, and Lean for verification. It takes as input an NL theorem+proof pair and produces the corresponding FL theorem+proof pair in Lean 4. The key insight behind our approach is to treat proof auto-formalization as *learning from demonstrations*: the LLM is guided not only by the NL proof but also by FL proofs retrieved using an NL/FL joint embedding model that leverages contrastive learning and encodes linear DAG traversals of Lean proofs. Rather than relying on randomly chosen few-shot examples, these retrieved proofs capture far richer *reusable formalization patterns* (tactic choices, DAG structures), providing grounded signals that guide generation toward Lean-verifiable proofs, as illustrated in Figure 1.

**Contributions:**

**The PROOFBRIDGE Auto-formalization Method and Tool:** We present PROOFBRIDGE, an LLM-based, retrieval-augmented proof auto-formalization framework. At its core is an *NL/FL joint embedding model* that maps semantically equivalent NL and FL theorem+proof pairs to nearby points in a shared space, enabling cross-modal retrieval of related FL proofs. We then fine-tune the SoTA LLM Kimina-Prover-RL-1.7B (Wang et al., 2025) to perform NL-to-Lean 4 proof translation, conditioned on the retrieved FL proofs and their relevance scores. During inference, generation is refined with an iterative verifier-guided repair loop combining Lean type-checking with LLM-based bi-directional equivalence proving to ensure syntactic correctness and semantic fidelity. (Section 4)

**NUMINAMATH-LEAN-PF Dataset:** We curate NUMINAMATH-LEAN-PF, a large-scale dataset of 38.9k NL↔Lean 4 theorem+proof pairs, specialized for *proof auto-formalization*. Each Lean theorem+proof pair is type-checked and paired with an NL counterpart. Additionally, we release MINIF2F-TEST-PF, a Lean v4.15.0 version of miniF2F-Test with 244 instances tailored for proof auto-formalization, enabling a consistent pipeline in the same Lean version. (Section 5.1)

**Extensive Experimental Evaluation:** Compared to the baseline encoder all-MiniLM-L6-v2, PROOFBRIDGE's cross-modal NL→FL retrieval achieves $3.28\times$ higher Recall Rate@$K$ at $K$=1 and $2.74\times$ MRR, with top-$K$ retrieved embeddings 23% closer and non-retrieved 104% farther. We evaluate PROOFBRIDGE against 13 SoTA LLMs, including foundation models (Gemini-2.5, GPT-5-mini) and automated proof synthesis LLMs (DeepSeek-Prover, STP, Leanabell-Prover, Kimina-Prover), using verifier-grounded metrics: *type correctness* (TC) and *semantic correctness* (SC, a new metric based on Lean bidirectional equivalence proofs). Built on Kimina-Prover-RL-1.7B, PROOFBRIDGE achieves +31.14% SC and +1.64% TC (pass@32) on MINIF2F-TEST-PF. (Section 5)

## 2  RELATED WORK

Our work lies at the intersection of three key AI-for-Math research areas: automated formal proof synthesis, auto-formalization, and retrieval-augmented learning for mathematical reasoning. We focus on the most relevant approaches and highlight differences from our unified framework.

**Auto-Formalization.** Auto-formalization translates NL mathematics into FL, but most existing work focuses on theorem formalization rather than proofs. ***Theorem-only approaches*** include Herald-translator (Gao et al., 2025), which extracts FL theorems from Mathlib4 and trains on informal counterparts, and Kimina-Autoformalizer (Wang et al., 2025), which fine-tunes models with expert iteration. These excel at translating theorems but cannot handle proofs. ***Proof auto-formalization*** has received limited attention. Draft-Sketch-Prove (Jiang et al., 2023) converts NL

proofs into formal sketches in Isabelle with open conjectures, then fills gaps using predefined tactics and tools like Sledgehammer (Paulsson & Blanchette, 2010). FormL4 (Lu et al., 2024) trains on GPT-4 informalized Mathlib proofs with process-supervised step-level Lean compilation feedback.

*Key Differences:* Our approach is the first to jointly learn representations for NL and FL theorem+proof pairs, enabling cross-modal retrieval to guide formalization. Unlike prior work, we leverage semantic relationships of NL and FL proofs for contextual guidance.

**Automated Formal Proof Synthesis (AFPS).** In this setting, given an FL theorem, the goal is to generate its FL proof. *Next-Tactic Prediction (NTP)* methods predict each proof step from the current proof state. Examples include GPT-f (Polu & Sutskever, 2020) for Metamath, LIsa (Jiang et al., 2021) for Isabelle, and PACT (Han et al., 2022) for Lean. They use tree search over proof states, prioritizing tactics by cumulative probability. While NTP ensures stepwise correctness via interactive theorem-prover verification, it suffers from long-horizon dependencies and repeated verification overhead. *Whole-Proof Generation (WPG)* methods generate complete FL proofs in a single pass, offering computational efficiency but risking cascading errors. Recent advances include DeepSeek-Prover-v1 (Xin et al., 2024a), which combines SFT with expert iteration, and TheoremLlama (Wang et al., 2024b), which improves in-context learning via curriculum-based training. DeepSeek-Prover-v2 (Ren et al., 2025) integrates NL reasoning with FL proof generation, while Kimina-Prover (Wang et al., 2025) applies reinforcement learning with compilation-based rewards (Jana et al., 2024).

*Key Differences:* Unlike AFPS approaches that assume FL theorems as input, our work addresses the more challenging task of generating theorem+proof pairs in FL from an NL input.

**Retrieval-Augmented Learning for Mathematics.** Recent work has explored retrieval-augmented mathematical reasoning, though not specifically for auto-formalization. TLAPS (Zhou, 2025) retrieves verified proofs to aid proof generation, while COPRA (Thakur et al., 2024) and REAL-Prover (Shen et al., 2025) retrieve lemmas or theorems to guide proof search. These methods rely on plain-text encoding. LeanSearch (Gao et al., 2024), HERALD (Gao et al., 2025), and RAutoformalizer (Liu et al., 2025) also use plain-text encoders for FL theorem retrieval; however, as shown in Section 5.5, this does not extend to the more demanding task of FL theorem+proof pair retrieval.

*Key Differences:* Our joint embedding enables NL/FL cross-modal retrieval of theorem+proof pairs and encodes the DAG structure of Lean proofs, unlike plain-text encoders. Capturing proof-structure semantics is essential for proof auto-formalization and is not addressed by existing tool-chains.

**Positioning our contributions.** The combination of joint embedding, retrieval augmentation, and unified translation distinguishes PROOFBRIDGE from prior work: *(a) Unified Proof Auto-Formalization:* We address complete translation (theorem + proof) rather than treating theorem formalization and proof synthesis separately. *(b) Joint Semantic Embedding:* Our contrastive learning framework for aligning NL and FL proofs is novel, enabling effective cross-modal retrieval. *(c) Retrieval-Augmented Translation:* We are the first to apply retrieval-augmented fine-tuning and generation to auto-formalization, leveraging semantic relationships of FL proofs to guide translation. *(d) Rigorous Evaluation:* We introduce systematic metrics for proof auto-formalization, covering type checking and semantic correctness via bi-directional equivalence proofs instead of proxy measures.

## 3 PRELIMINARIES: TACTIC-STYLE PROOFS IN LEAN

Lean (Moura & Ullrich, 2021) is a functional programming language and interactive theorem prover that is based on the propositions-as-types principle, where proving a proposition is equivalent to constructing a term of the corresponding type. Rather than building these terms manually, users write proofs in a tactic language, which provides high-level steps to guide term construction. Lean 4 (henceforth Lean) represents tactic-style proofs as directed acyclic graphs (DAGs) of *proof states* and *tactics*, automatically generating the corresponding proof term in the background. The kernel then verifies the term, ensuring correctness by enforcing the axiomatic foundations of Lean's logic, the Calculus of Inductive Constructions. This combination of a formal system and a small, trusted kernel provides strong confidence in the validity of proofs. In the DAG (Figure 2) of a Lean proof:

- Each **proof state** $S_i \equiv [G_1, \cdots, G_n]$ consists of a sequence of zero or more *open goals*. Initial state $S_0$ has one goal, the theorem $T_{FL} \equiv pr \vdash cn$ itself. Leaf-level states have no open goal.
- Each **open goal** $G_i \equiv pr_i \vdash cn_i$ of a proof state represents a proposition $cn_i$ that needs to be proven, given a set of premises $pr_i$.

- Each **tactic** $tac_i$ represents a proof step. It is a high-level command (rooted in metaprogramming) applied to an open goal $G_i$, producing a new proof state. If the resulting proof state has no open goal, it directly resolves the current goal. A parent goal is resolved once all subgoals are resolved.

Tactic-style proof synthesis in Lean follows a *sequential decision process*. Lean provides an interactive REPL (Leanprover, 2025) that applies tactics step by step to manipulate proof states. An FL proof is a sequence of tactics, and at each step, the REPL updates the proof state if the tactic is valid or returns an error identifying the faulty one. Each tactic advances the proof by breaking the current goal into simpler subgoals, similar to the '*suffices to show*' construct.

**Proof Auto-formalization as a Learning Problem.** Given an NL theorem+proof pair $M_{\text{NL}} = \langle T_{\text{NL}}, P_{\text{NL}} \rangle$, the goal is to learn a function $f : M_{\text{NL}} \mapsto M_{\text{FL}}$ that produces a corresponding Lean theorem+proof pair $M_{\text{FL}} = \langle T_{\text{FL}}, P_{\text{FL}} \rangle$. Here, $T_{\text{FL}} \equiv pr \vdash cn$ denotes the formal theorem, and $P_{\text{FL}} \equiv (\texttt{tac}_0, \ldots, \texttt{tac}_{n-1})$ is the proof as a sequence of tactics. The generated pair must satisfy:

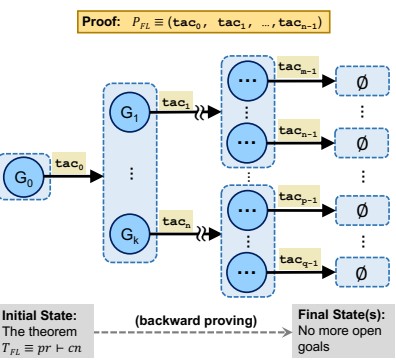

Figure 2: **Tactic-style Proof.** A Lean proof represented as a DAG of tactics.

(a) *Type correctness:* $M_{\text{FL}} = \langle T_{\text{FL}}, P_{\text{FL}} \rangle$ passes Lean type-checking, ensuring that $P_{\text{FL}}$ proves $T_{\text{FL}}$ with no open goals in the DAG.
(b) *Semantic correctness:* FL theorem is semantically equivalent to the NL one, i.e., $T_{\text{FL}} \equiv T_{\text{NL}}$.

# 4 OUR APPROACH AND TOOL ARCHITECTURE

## 4.1 JOINT EMBEDDING OF NL AND LEAN PROOFS FOR CROSS-MODAL RETRIEVAL

A core component of our framework is the *joint embedding model*, which learns to represent NL theorem+proof pairs and their FL (Lean) counterparts in a shared semantic space. The goal is to align these modalities so that cross-modal retrieval between NL and FL becomes possible. Let $M_{\text{FL}}^{(i)} = \langle T_{\text{FL}}^{(i)}, P_{\text{FL}}^{(i)} \rangle$ denote an FL theorem+proof pair, and let $\mathcal{D} = \{M_{\text{FL}}^{(i)}\}_{i=1}^N$ be a large database of such pairs. Given an NL theorem+proof pair $M_{\text{NL}}$, the model retrieves a subset $\mathcal{R}(M_{\text{NL}}, \mathcal{D}) \subset \mathcal{D}$ of size $K \ll |\mathcal{D}|$ that serves as in-context demonstrations to guide downstream auto-formalization.

During training, $M_{\text{NL}}^{(i)}$ and $M_{\text{FL}}^{(i)}$ are encoded into vectors by modality-specific encoders. Each encoder is initialized with a pre-trained model, and a subset of parameters is subsequently fine-tuned. The NL encoder $f$ maps $M_{\text{NL}}^{(i)}$ to a vector $v_{\text{NL}}^{(i)} = f(M_{\text{NL}}^{(i)}, \theta_f \| \phi_f) \in \mathbb{R}^d$, and the FL encoder $g$ maps $M_{\text{FL}}^{(i)}$ to $v_{\text{FL}}^{(i)} = g(M_{\text{FL}}^{(i)}, \theta_g \| \phi_g) \in \mathbb{R}^d$, where $\theta$ and $\phi$ denote frozen and trainable parameters, respectively, and $d$ is the shared embedding dimension. The details of each encoder are as follows:

- **NL encoder $f(M_{\text{NL}}^{(i)}, \theta_f \| \phi_f)$:** To encode $M_{\text{NL}}^{(i)}$, we use `all-MiniLM-L6-v2` (Reimers & Gurevych, 2019), a lightweight model (22.7M parameters) that effectively captures semantic similarity in NL. It encodes $M_{\text{NL}}^{(i)}$ into 384-dimensional embeddings, thereby projected into the joint embedding space of dimension $d = 512$ via a linear layer included in the trainable set $\phi_f$.

- **FL encoder $g(M_{\text{FL}}^{(i)}, \theta_g \| \phi_g)$:** Given $M_{\text{FL}}^{(i)} = \langle T_{\text{FL}}^{(i)}, P_{\text{FL}}^{(i)} \rangle$, we first extract the linearized DAG traversal of tactics from $P_{\text{FL}}^{(i)}$ using Lean REPL (Leanprover, 2025). This traversal is represented as an ordered sequence of proof-state transformations induced by successive tactic applications: $S_0 \xrightarrow{\texttt{tac}_0} S_1 \xrightarrow{\texttt{tac}_1} \cdots \xrightarrow{\texttt{tac}_{H-1}} S_H$, where $S_0 \equiv T_{\text{FL}}^{(i)}$, each $S_h \equiv [G_1, \ldots, G_l]$ denotes a proof state consisting of zero or more open goals, and $\texttt{tac}_{h-1}$ is the tactic applied at step $h$. This sequence captures the entire proof as an ordered series of state transformations. To create embeddings for the full proof, we first encode each state $S_h$ using LeanDojo's `ByT5` proof-state encoder (Yang et al., 2023) (218M parameters), producing embeddings of size 1,472 per state. We then obtain a single embedding for the entire proof via mean-pooling, which is subsequently projected into a shared semantic space of dimension $d = 512$ using a linear layer included in the trainable parameters $\phi_g$. The intuition behind this approach is to ensure that semantically similar proofs (those with similar DAG structures of proof states and tactics) produce similar embeddings.

**Contrastive Learning.** To enable cross-modal retrieval between NL and FL representations, it is essential to align the two modalities in the embedding space. Specifically, for each positive pair $\big(M_{\text{NL}}^{(i)}, M_{\text{FL}}^{(i)}\big)$, we aim for their embeddings $\big(v_{\text{NL}}^{(i)}, v_{\text{FL}}^{(i)}\big)$ to exhibit high cosine similarity, while embeddings of mismatched pairs are pushed apart. Denoting $\ell_2$-normalization by $\widehat{v} = v/\|v\|_2$ and defining the cosine similarity between two embeddings $u$ and $w$ as $[\widehat{u}, \widehat{w}]$, we adopt the following symmetric contrastive loss for a mini-batch $\mathcal{B} = \big\{\big(M_{\text{NL}}^{(i)}, M_{\text{FL}}^{(i)}\big)\big\}_{i=1}^{n}$ of NL and FL pairs:

$$\mathcal{L}(\mathcal{B}) = -\frac{1}{2n} \sum_{i=1}^{n} \left[ \log\left( \frac{\exp\big([\widehat{v}_{\text{NL}}^{(i)}, \widehat{v}_{\text{FL}}^{(i)}]/\tau\big)}{\sum_{j=1}^{n} \exp\big([\widehat{v}_{\text{NL}}^{(i)}, \widehat{v}_{\text{FL}}^{(j)}]/\tau\big)} \right) + \log\left( \frac{\exp\big([\widehat{v}_{\text{FL}}^{(i)}, \widehat{v}_{\text{NL}}^{(i)}]/\tau\big)}{\sum_{j=1}^{n} \exp\big([\widehat{v}_{\text{FL}}^{(i)}, \widehat{v}_{\text{NL}}^{(j)}]/\tau\big)} \right) \right] \quad (1)$$

where $\tau > 0$ is a temperature hyperparameter. This loss encourages each NL embedding to be closest to its corresponding FL embedding, and vice versa, using other in-batch embeddings as negatives. The negatives are sampled randomly for each mini-batch.

**NL/FL Cross-Modal Retrieval.** We precompute the normalized embeddings $\big\{\widehat{v}_{\text{NL}}^{(i)}\big\}_{i=1}^{|\mathcal{D}|}$ and $\big\{\widehat{v}_{\text{FL}}^{(j)}\big\}_{j=1}^{|\mathcal{D}|}$ for all NL and FL theorem+proof pairs respectively in our database $\mathcal{D}$, which enables efficient cross-modal retrieval. Given a query theorem+proof pair in either source modality (NL or FL), we encode it into the shared semantic space (yielding $\widehat{q}_{\text{NL}}$ or $\widehat{q}_{\text{FL}}$) and compute cosine similarities with all items in the target modality, producing the set $\big\{[\widehat{q}_{\text{NL}}, \widehat{v}_{\text{FL}}^{(j)}]\big\}_{j=1}^{|\mathcal{D}|}$ or $\big\{[\widehat{q}_{\text{FL}}, \widehat{v}_{\text{NL}}^{(i)}]\big\}_{i=1}^{|\mathcal{D}|}$, depending on the retrieval direction. The top-$K$ nearest neighbors from these sets are then selected as demonstrations, reflecting similar proof structures, patterns, and mathematical domains.

## 4.2 Retrieval-Augmented Fine-Tuning for Proof Auto-Formalization

We fine-tune an LLM to translate NL theorem+proof pairs into FL (Lean), conditioned on retrieved FL demonstrations that provide rich contextual knowledge. For each training instance, we construct a prompt containing (a) input NL theorem+proof pair $M_{\text{NL}}$ and (b) top-$K$ retrieved FL theorem+proof pairs: $\mathcal{R}(M_{\text{NL}}, \mathcal{D}) = \{M_{\text{FL}}^{(k)}\}_{k=1}^{K}$ with relevance scores $\{r^{(k)}\}_{k=1}^{K}$. The retrieved examples demonstrate how similar mathematical concepts and proof strategies are formalized in Lean. We include relevance scores to help the model weight the importance of each retrieved example.

**Training Objective.** We fine-tune Kimina-Prover-RL-1.7B (Wang et al., 2025) using supervised learning on our NUMINAMATH-LEAN-PF dataset (details in Section 5.1). The model is trained to generate an FL theorem+proof pair $\widetilde{M}_{\text{FL}}$ given the input context. This retrieval-augmented approach allows the LLM to learn from similar formalization patterns rather than generating formal theorems in isolation. As illustrated in Figure 1b, we use the standard auto-regressive language modeling loss:

$$\mathcal{L}_{\text{CE}} = -\frac{1}{|\mathcal{T}|} \sum_{t=1}^{|\mathcal{T}|} \log P_\theta\left(\tau_t \mid \tau_{<t}, \mathcal{C}\right) \quad (2)$$

where $\mathcal{T} = \widetilde{M}_{\text{FL}}$ is the generated formalization tokenized as sequence $(\tau_1, \ldots, \tau_{|\mathcal{T}|})$, $\mathcal{C}$ represents the input context (NL theorem+proof and retrieved FL examples), and $\theta$ are the LLM parameters. This corresponds to the cross-entropy loss between $\widetilde{M}_{\text{FL}}$ and the gold-standard formalization $M_{\text{FL}}$.

## 4.3 Iterative Proof Repair with Verifier Feedback

During inference, we perform retrieval-augmented proof auto-formalization with the fine-tuned LLM (Figure 1c). However, LLM being a stochastic model may still generate FL theorem+proof pair that contain syntactic errors or semantic misalignments with the input NL theorem+proof. To address, we implement an iterative repair mechanism that combines Lean's type checker with semantic equivalence verification. For an input NL theorem+proof pair $M_{\text{NL}} = \langle T_{\text{NL}}, P_{\text{NL}} \rangle$ the LLM generates an FL counterpart $\widetilde{M}_{\text{FL}} = \langle \widetilde{T}_{\text{FL}}, \widetilde{P}_{\text{FL}} \rangle$, on which we perform two types of verification:

1. **Syntactic Verification:** We compile $\widetilde{M}_{\text{FL}}$ using Lean's type checker. If compilation fails, we extract the specific error message and location from Lean's diagnostic output.
2. **Semantic Verification:** We assess whether the generated theorem $\widetilde{T}_{\text{FL}}$ accurately represents the original NL theorem $T_{\text{NL}}$ using an LLM-based equivalence judge.

**Repair Process.** When either syntactic or semantic verification fails, we initiate an iterative repair process. The procedure terminates once both checks succeed or the maximum number of repair attempts ($R_{\max} = 5$) is reached. This bounded, iterative strategy improves the reliability of proof auto-formalization by catching and correcting common errors while maintaining computational efficiency. The overall process is described in Algorithm 1.

---

**Algorithm 1** Iterative Proof Repair

---

1: **Input:** NL theorem+proof pair $M_{\mathrm{NL}} = \langle T_{\mathrm{NL}}, P_{\mathrm{NL}} \rangle$,
       Initial FL theorem+proof pair $\widetilde{M}_{\mathrm{FL}}^{(0)} = \langle \widetilde{T}_{\mathrm{FL}}^{(0)}, \widetilde{P}_{\mathrm{FL}}^{(0)} \rangle$
2: **Output:** Verified FL theorem+proof pair or FAILURE
3: **for** $i = 0$ to $R_{\max} - 1$ **do**
4:     syntaxOK $\leftarrow$ LeanTypeCheck$\left(\widetilde{M}_{\mathrm{FL}}^{(i)}\right)$
5:     semanticsOK $\leftarrow$ SemanticEquivalence$\left(T_{\mathrm{NL}}, \widetilde{T}_{\mathrm{FL}}^{(i)}\right)$
6:     **if** syntaxOK $\wedge$ semanticsOK **then return** $\widetilde{M}_{\mathrm{FL}}^{(i)}$
7:     feedback $\leftarrow$ GenerateFeedback(syntaxOK, semanticsOK)
8:     $\widetilde{M}_{\mathrm{FL}}^{(i+1)} \leftarrow$ LLMRepair(feedback)
9: **end for**
10: **return** FAILURE

---

## 5 EXPERIMENTAL EVALUATION

### 5.1 DATASETS AND PREPARATION: NUMINAMATH-LEAN-PF AND MINIF2F-TEST-PF

For training, we construct NUMINAMATH-LEAN-PF from NuminaMath-LEAN (Wang et al., 2025), containing 104,155 competition-level problems in algebra, geometry, number theory, combinatorics, and calculus. Each instance pairs an NL theorem $T_{\mathrm{NL}}$ with a human-written Lean v4.15.0 theorem $T_{\mathrm{FL}}$; 38,951 include FL proofs $P_{\mathrm{FL}}$ (30% human-written, rest by KiminaProver), forming $\{T_{\mathrm{NL}}, \langle T_{\mathrm{FL}}, P_{\mathrm{FL}} \rangle\}$. Next, we prepare NUMINAMATH-LEAN-PF via the following steps:

**Formal Verification and Repair.** Each $\langle T_{\mathrm{FL}}, P_{\mathrm{FL}} \rangle$ is type-checked using Lean REPL (Leanprover, 2025). About 6% (2,337) failed due to syntax, library mismatches, or incomplete proofs. These were automatically repaired via Gemini-2.5-Pro: error messages and locations are extracted from Lean, used to prompt the LLM for corrections, and re-verified iteratively up to five times.

**NL Proof Generation.** As NuminaMath-LEAN provides only theorems, we generate NL proofs in two stages. First, *solution sketch retrieval* searches for $T_{\mathrm{NL}}$ in NuminaMath 1.5 (Li et al., 2024) (896k problem-solution pairs), retrieving NL solution sketches (median 79 words) for 25,792 instances (66%). Next, *FL-to-NL informalization* uses Gemini-2.5-Pro to translate $P_{\mathrm{FL}}$ into NL proofs, using available sketches, yielding 38,951 NL-FL theorem+proof pairs $\{\langle T_{\mathrm{NL}}, P_{\mathrm{NL}} \rangle, \langle T_{\mathrm{FL}}, P_{\mathrm{FL}} \rangle\}$.

For validation, we curate MINIF2F-TEST-PF by combining two versions of miniF2F-test (Zheng et al., 2022), a widely-used auto-formalization benchmark. It contains 244 Olympiad-level problems from the AIME, AMC, IMO, and undergraduate courses in algebra, number theory, and inequalities. We use the Lean v4.15.0 version (NuminaMath, 2025) and add missing NL proofs from Yang (2025).

### 5.2 EVALUATION METRICS

**Metrics for NL/FL Cross-Modal Retrieval.** We evaluate cross-modal alignment of our joint embedding model in two directions. NL $\rightarrow$ FL measures retrieval of FL theorem+proof pairs given an NL input, which is relevant for proof auto-formalization, while FL $\rightarrow$ NL assesses the reverse. For a test pair $(M_{\mathrm{NL}}, M_{\mathrm{FL}})$, a retrieval in the NL $\rightarrow$ FL direction is deemed successful if the model retrieves the FL counterpart $M_{\mathrm{FL}}$ given $M_{\mathrm{NL}}$, and unsuccessful otherwise; the FL $\rightarrow$ NL direction is evaluated analogously. We assess retrieval performance using five metrics. *(i) Recall Rate @ K* measures the percentage of queries for which the query's cross-modal counterpart appears among the top-$K$ retrieved results. We report $K = 1, 5, 10, 20, 50$. *(ii) Mean Reciprocal Rank (MRR)* is the average reciprocal rank of the retrieved cross-modal counterpart for each query, $\mathrm{MRR} = \frac{1}{N} \sum_{q=1}^{N} \frac{1}{\mathrm{rank}_q}$, indicating how highly it is ranked. *(iii) Cosine Similarity of Top-K Retrieved* measures the cosine similarity between the query embedding and those of the top-$K$ retrieved instances. For each query, we sort these scores in ascending order and record three statistics: median (M), $25^{\mathrm{th}}$ percentile (Q1), and $75^{\mathrm{th}}$ percentile (Q3), and report their average over the test set. *(iv) Cosine Similarity of Non-Retrieved* applies the same procedure to all non-retrieved instances and reports the median (M), $25^{\mathrm{th}}$ percentile (Q1), and $75^{\mathrm{th}}$ percentile (Q3) averaged over the test set. *(v) mean Median Gap (mMG)* measures the difference between the median (M) cosine similarity of top-$K$ retrieved and that of non-retrieved instances, averaged over $K = 1, 5, 10, 20, 50$.

**Metrics for Proof Auto-Formalization.** Given $M_{\mathrm{NL}} = \langle T_{\mathrm{NL}}, P_{\mathrm{NL}} \rangle$, a proof auto-formalizer generates an FL version $\widetilde{M}_{\mathrm{FL}} = \langle \widetilde{T}_{\mathrm{FL}}, \widetilde{P}_{\mathrm{FL}} \rangle$. We evaluate this using two metrics: *(i) Type Correctness (TC)* checks if $\widetilde{M}_{\mathrm{FL}}$ passes Lean's type-checker, i.e., $\widetilde{P}_{\mathrm{FL}}$ proves $\widetilde{T}_{\mathrm{FL}}$ without using keyword `sorry`.

*(ii) Semantic Correctness (SC)* is measured only for type-correct outputs and checks whether $\widetilde{T}_{\mathrm{FL}}$ is definitionally equal to $T_{\mathrm{FL}}$ (we admit some propositional equalities) by prompting Gemini-2.5-Pro up to five times to produce a Lean proof of the bi-directional equivalence $\widetilde{T}_{\mathrm{FL}} \leftrightarrow T_{\mathrm{FL}}$ using a restricted set of tactics, e.g., `rfl`, `simp`, `ring`, etc. (see Appendix A.4 for details). Although relying on an LLM judge, correctness is determined via the Lean proof, not the LLM's judgment alone. TC and SC are reported as *pass@k*, i.e., over the top-$k$ generated candidates, for $k = 1, 2, 4, 8, 16, 32$.

## 5.3 STATE-OF-THE-ART BASELINES

**SoTA for NL/FL Cross-Modal Retrieval.** To our knowledge, no existing model jointly embeds theorems and proofs in NL and FL. Pre-trained encoders alone do not yield embeddings suitable for meaningful cross-modal retrieval. To illustrate, we evaluate two *SoTA Encoders*: Qwen3-Embedding-8B (Zhang et al., 2025b) and E5-Mistral-7B-Instruct (Wang et al., 2024a). Qwen3 allows user-defined output dimensions up to 4096, and we use 512 to match our joint embedding model, while E5-Mistral (used by LeanSearch-PS (Shen et al., 2025)) has a fixed dimension of 4096. We also include a *Baseline Encoder*, all-MiniLM-L6-v2 (Reimers & Gurevych, 2019), used for the NL encoder in PROOFBRIDGE, producing 384-dim embeddings. All these encoders treat theorem+proof pairs as plain text, ignoring the DAG structure of FL proofs, which PROOFBRIDGE explicitly leverages. Retrieval is performed via cosine similarity in the respective embedding spaces.

**SoTA Tools for Proof Auto-Formalization.** Existing proof auto-formalization tools include DSP (Jiang et al., 2023), which supports only Isabelle, making comparison infeasible since we target Lean 4. Another recent tool, FormL4 (Lu et al., 2024), has not yet released its trained model. We therefore compare three categories of tools. First, we consider four *foundation models*, including GPT-5-mini (OpenAI, 2025) and the Gemini-2.5 variants, Flash-Lite, Flash, and Pro (Comanici et al., 2025). Next, we evaluate seven *AFPS LLMs* that generate FL proofs with tactics: DeepSeek-Prover-V1.5-RL (Xin et al., 2024b), STP_model_Lean_0320 (Dong & Ma, 2025), Goedel-Prover-SFT (Lin et al., 2025), Leanabell-Prover-V2-KM (Zhang et al., 2025a), and three Kimina-Prover variants (Wang et al., 2025) (RL-1.7B, Distill-8B, 72B). Finally, we evaluate two *auto-formalization LLMs*: Kimina-Autoformalizer-7B (Wang et al., 2025) and Herald-Translator (Gao et al., 2025).

## 5.4 TRAINING AND EVALUATION SETUP

We train our joint embedding model on 90% (35,056 instances) of NUMINAMATH-LEAN-PF and evaluate it on the remaining 3,895 instances. The split is domain-stratified across all mathematical areas, ensuring hard negatives in the test set. The train split serves as a database $\mathcal{D}$ of FL theorem+proof pairs. PROOFBRIDGE, built on Kimina-Prover-RL-1.7B, is SFT-tuned for NL-to-FL translation using $(M_{\mathrm{NL}}, M_{\mathrm{FL}})$ from NUMINAMATH-LEAN-PF, with the joint embedding model retrieving the top-5 relevant FL proofs from $\mathcal{D}$ for retrieval-augmented SFT and inference. Inference-time iterative proof repair is applied, and the model is evaluated on MINIF2F-TEST-PF. See Appendices A.2–A.3 for implementation and training details, and Appendix A.5 for an example inference.

The *SoTA Tools* are evaluated in three settings: (a) *zero-shot*, with no in-context I/O examples; (b) *random few-shot*, with five randomly selected in-context examples; and (c) *text-based retrieval few-shot*, where the top-5 FL theorem+proof pairs are retrieved from $\mathcal{D}$ via Qwen3-Embedding-8B and paired with their NL counterparts as in-context examples. Further, we evaluate a *SoTA Two-Step* setting: a theorem-only auto-formalizer (T1) first translates $T_{\mathrm{NL}}$ to $\widetilde{T}_{\mathrm{FL}}$, and an AFPS LLM (T2) then generates $\langle \widetilde{T}_{\mathrm{FL}}, \widetilde{P}_{\mathrm{FL}} \rangle$ from $\widetilde{T}_{\mathrm{FL}}$, both in zero-shot. For *pass@k*, we sample the top-$k$ from T1, select one that is TC and judged equivalent to $T_{\mathrm{NL}}$ by Gemini-2.5-Pro (SC is not used as the gold $T_{\mathrm{FL}}$ is withheld until the pipeline finishes), and then generate the top-$k$ proof candidates from T2.

## 5.5 EXPERIMENTAL RESULTS

**NL/FL Cross-Modal Retrieval.** Table 1 compares PROOFBRIDGE's joint embedding model with the two *SoTA Encoders* and the *Baseline Encoder*. Since PROOFBRIDGE is obtained by contrastively training the NL encoder together with an FL encoder, all improvements are reported relative to the original NL encoder (the Baseline Encoder). PROOFBRIDGE achieves consistently higher recall rates across all top-$K$ values: for NL→FL, it yields $3.28\times$ gain for NL→FL and $1.94\times$ for FL→NL at $K = 1$. MRR also improves by $2.74\times$ and $1.79\times$ for NL→FL and FL→NL, respectively. This

Table 1: **NL/FL Cross-Modal Retrieval Performance.** Retrieval performance of *SoTA* and *Baseline* encoders versus PROOFBRIDGE's joint embedding. (Q1: 25$^{th}$ %tile, M: median, Q3: 75$^{th}$ %tile)

| Method | Retrieval Direction | Recall Rate @ K (%) ↑ | | | | | MRR ↑ | | Cos. Similarity of top-K Retrieved ↑ | | | | | Cos. Similarity of NOT Retrieved ↓ | | | | | Gap ↑ (mMG) |
|---|---|---|---|---|---|---|---|---|---|---|---|---|---|---|---|---|---|---|---|
| | | K=1 | K=5 | K=10 | K=20 | K=50 | | | K=1 | K=5 | K=10 | K=20 | K=50 | K=1 | K=5 | K=10 | K=20 | K=50 | |
| Qwen3-Embedding-8B (*SoTA Encoder*, 8B params) | NL → FL | 46.75 | 71.96 | 82.49 | 87.04 | 93.34 | 0.567 | Q1: M: Q3: | 0.74 0.74 0.74 | 0.67 0.68 0.70 | 0.62 0.63 0.66 | 0.60 0.62 0.63 | 0.57 0.58 0.60 | 0.317 0.362 0.410 | 0.317 0.362 0.410 | 0.317 0.362 0.410 | 0.317 0.362 0.409 | 0.317 0.362 0.409 | 0.29 |
| | FL → NL | 44.68 | 68.26 | 79.79 | 83.78 | 87.36 | 0.506 | Q1: M: Q3: | 0.76 0.76 0.76 | 0.66 0.67 0.68 | 0.63 0.63 0.65 | 0.62 0.63 0.64 | 0.59 0.60 0.62 | 0.309 0.362 0.418 | 0.309 0.362 0.418 | 0.309 0.362 0.418 | 0.309 0.362 0.418 | 0.308 0.362 0.417 | 0.30 |
| E5-Mistral-7B-Instruct (*SoTA Encoder*, 7B params) | NL → FL | 35.60 | 53.22 | 60.22 | 67.82 | 77.18 | 0.441 | Q1: M: Q3: | 0.86 **0.86** 0.86 | 0.83 **0.83** 0.84 | 0.82 **0.83** 0.83 | 0.82 **0.82** 0.83 | 0.81 **0.81** 0.82 | 0.711 0.730 0.749 | 0.711 0.730 0.749 | 0.711 0.730 0.749 | 0.711 0.730 0.749 | 0.710 0.730 0.749 | 0.10 |
| | FL → NL | 30.27 | 41.93 | 46.41 | 50.83 | 57.88 | 0.359 | Q1: M: Q3: | 0.87 **0.87** 0.87 | 0.85 **0.85** 0.86 | 0.84 **0.85** 0.85 | 0.84 **0.84** 0.85 | 0.83 **0.83** 0.84 | 0.704 0.735 0.760 | 0.704 0.735 0.760 | 0.704 0.735 0.760 | 0.704 0.735 0.760 | 0.704 0.734 0.759 | 0.11 |
| all-MiniLM-L6-v2 (*Baseline Encoder*, 22.7M params) | NL → FL | 16.06 | 30.93 | 38.63 | 47.31 | 60.95 | 0.237 | Q1: M: Q3: | 0.60 0.60 0.60 | 0.52 0.54 0.55 | 0.50 0.51 0.53 | 0.47 0.49 0.51 | 0.44 0.45 0.58 | 0.147 0.210 0.274 | 0.147 0.210 0.274 | 0.147 0.210 0.273 | 0.147 0.209 0.273 | 0.146 0.208 0.271 | 0.31 |
| | FL → NL | 26.35 | 45.12 | 54.28 | 63.75 | 75.54 | 0.355 | Q1: M: Q3: | 0.58 0.58 0.58 | 0.52 0.53 0.54 | 0.50 0.51 0.53 | 0.47 0.49 0.51 | 0.44 0.46 0.48 | 0.142 0.208 0.278 | 0.142 0.208 0.278 | 0.142 0.208 0.277 | 0.142 0.207 0.277 | 0.141 0.206 0.275 | 0.31 |
| **PROOFBRIDGE** (*Proposed*, 22.7M + 218M + 1M params) | NL → FL | **52.83** | **79.81** | **87.06** | **91.49** | **95.08** | **0.650** | Q1: M: Q3: | 0.76 0.76 0.76 | 0.66 0.68 0.71 | 0.62 0.64 0.68 | 0.57 0.60 0.64 | 0.49 0.52 0.58 | -0.134 **-0.009** 0.123 | -0.135 **-0.010** 0.123 | -0.135 **-0.010** 0.122 | -0.135 **-0.010** 0.121 | -0.136 **-0.012** 0.117 | **0.65** |
| | FL → NL | **51.23** | **78.77** | **86.18** | **90.50** | **94.83** | **0.635** | Q1: M: Q3: | 0.77 0.77 0.77 | 0.67 0.69 0.71 | 0.62 0.64 0.68 | 0.57 0.60 0.64 | 0.49 0.52 0.58 | -0.135 **-0.009** 0.124 | -0.135 **-0.009** 0.123 | -0.135 **-0.009** 0.122 | -0.135 **-0.011** 0.121 | -0.136 **-0.012** 0.117 | **0.65** |

indicates that PROOFBRIDGE more frequently retrieves the correct cross-modal counterpart among the highest-ranked results. For the NL and FL embeddings by PROOFBRIDGE, the median (M) cosine similarity of retrieved cross-modal instances averages $0.64$ across top-$K$ ($K = 1, 5, 10, 20, 50$) in both directions, showing that select cross-modal theorem+proof pairs are tightly clustered. In contrast, non-retrieved instances are much farther apart, averaging $-0.01$. Compared to the Baseline, retrieved-pair similarities increase by $23\%$, while non-retrieved similarities decrease by $104\%$.

PROOFBRIDGE outperforms the *SoTA Encoders* in recall and MRR, with a much higher mMG despite being $32\times$ smaller, showing a clearer separation between retrieved and non-retrieved items. For NL→FL, E5-Mistral-7B-Instruct attains an mMG of $0.10$, while Qwen3-Embedding-8B reaches $0.29$. We believe these low values arise because: *first*, these QA-oriented encoders capture coarse domain-level signals rather than fine-grained mathematical semantics, so most mathematical texts cluster together; *second*, as plain-text, non-DAG-aware encoders, they rely on superficial lexical cues (keywords), but in Lean many proofs share the same tactics, making keyword overlap non-discriminative. In contrast, PROOFBRIDGE leverages the DAG to distinguish proofs, achieving an mMG of $0.65$. Overall, encoding Lean proofs via linearized DAG traversals and contrastive alignment with a DAG-aware FL encoder yield an effective joint embedding space, where equivalent NL-FL pairs cluster tightly and inequivalent ones remain well separated. This enables reliable retrieval of the most relevant FL demonstrations to condition the LLM during auto-formalization.

**Proof Auto-Formalization.** Table 2 reports the proof auto-formalization performance of 13 SoTA LLM-based tools. *Theorem auto-formalization LLMs* achieve 0% TC and SC across all pass@k. These models are designed to formalize theorem statements only, leaving proofs as `sorry`. They lack knowledge of proof DAGs and tactics, making them unsuitable for end-to-end proof auto-formalization. Among *foundation models*, Gemini-2.5-Flash-Lite achieves 2.87% SC and 21.31% TC at pass@32, which increase to 4.10% SC, 18.44% TC for Flash and 8.61% SC, 31.56% TC for Pro. GPT-5-mini attains 9.02% TC and 34.84% SC at pass@32. They struggle with the strict syntax and semantics of specialized FLs like Lean, which are underrepresented in their training data. The *SoTA Two-Step* achieves 43.44% SC and 59.43% TC, but it is prone to cascading errors: an incorrect FL theorem from the first model causes the second to produce a semantically incorrect theorem+proof pair. Among the *SoTA AFPS LLMs*, Kimina-Prover-72B achieves the strongest zero-shot performance at pass@32, with 46.31% SC and 79.51% TC. We build PROOFBRIDGE on top of a smaller variant, Kimina-Prover-RL-1.7B, by retrieving five relevant FL proofs via NL/FL cross-modal retrieval and using them for retrieval-augmented SFT and inference, along with iterative proof repair. PROOFBRIDGE surpasses the zero-shot performance of Kimina-Prover-RL-1.7B by +22.54% SC and +20.49% TC, and its random few-shot performance by +31.14% SC and +1.64% TC.

In Figure 3, we present the pass@32 performance across mathematical domains. Following the taxonomy by Zheng et al. (2022), the benchmark includes 6 induction/sequence (2.46%), 69 number-theory (28.28%), 90 algebra (36.89%), and 79 contest problems (32.38%) sourced from AIME, AMC, and IMO. PROOFBRIDGE performs best on number theory, achieving over 85% SC, while contest problems remain the most challenging, reaching only about 35% SC.

Table 2: **Proof Auto-Formalization Performance.** Comparison of LLM-based tools on *Semantic Correctness (SC)* and *Type Correctness (TC)* across pass@$k$ metrics ($k \in \{1, 2, 4, 8, 16, 32\}$).

| Setting | LLM/Tool | Semantic Correctness (SC) (%) ↑ | | | | | | Type Correctness (TC) (%) ↑ | | | | | |
|---|---|---|---|---|---|---|---|---|---|---|---|---|---|
| | | pass@1 | pass@2 | pass@4 | pass@8 | pass@16 | pass@32 | pass@1 | pass@2 | pass@4 | pass@8 | pass@16 | pass@32 |
| *SoTA Tools* (zero-shot) | Kimina-Prover-RL-1.7B | 9.02 | 13.93 | 22.54 | 30.33 | 35.25 | 40.16 | 26.23 | 41.80 | 56.15 | 62.70 | 68.03 | 75.00 |
| | Kimina-Prover-Distill-8B | 10.66 | 18.85 | 23.77 | 32.38 | 37.70 | 41.80 | 27.05 | 43.03 | 58.20 | 63.52 | 72.95 | 75.82 |
| | Kimina-Prover-72B | 12.70 | 21.31 | 25.00 | 34.84 | 38.52 | 43.03 | 30.33 | 45.08 | 61.07 | 69.26 | 75.41 | 79.51 |
| *SoTA Tools* (random few-shot) | Kimina-Autoformalizer-7B | 0.00 | 0.00 | 0.00 | 0.00 | 0.00 | 0.00 | 0.00 | 0.00 | 0.00 | 0.00 | 0.00 | 0.00 |
| | Herald-Translator | 0.00 | 0.00 | 0.00 | 0.00 | 0.00 | 0.00 | 0.00 | 0.00 | 0.00 | 0.00 | 0.00 | 0.00 |
| | Gemini-2.5-Flash-Lite | 0.00 | 0.00 | 0.00 | 0.00 | 1.23 | 2.87 | 0.82 | 4.51 | 9.02 | 13.93 | 18.03 | 21.31 |
| | Gemini-2.5-Flash | 0.00 | 0.82 | 2.05 | 2.86 | 3.28 | 4.10 | 2.45 | 4.92 | 7.38 | 12.30 | 15.98 | 18.44 |
| | Gemini-2.5-Pro | 1.23 | 1.23 | 3.28 | 4.92 | 6.97 | 8.61 | 9.84 | 13.52 | 18.85 | 23.77 | 29.10 | 31.56 |
| | GPT-5-mini | 0.41 | 1.23 | 4.51 | 6.97 | 7.38 | 9.02 | 4.92 | 9.43 | 20.08 | 28.28 | 32.38 | 34.84 |
| | DeepSeek-Prover-V1.5-RL | 3.69 | 6.15 | 8.61 | 9.02 | 11.07 | 12.30 | 8.20 | 14.34 | 19.67 | 24.18 | 28.68 | 35.66 |
| | STP_model_Lean_0320 | 4.51 | 6.56 | 8.20 | 9.84 | 11.48 | 13.11 | 12.70 | 18.03 | 23.36 | 28.69 | 33.61 | 39.34 |
| | Goedel-Prover-SFT | 4.92 | 5.33 | 7.38 | 8.20 | 12.70 | 16.39 | 13.52 | 17.21 | 25.41 | 31.56 | 36.88 | 42.21 |
| | Leanabell-Prover-V2-KM | 6.97 | 9.43 | 10.66 | 13.52 | 15.16 | 18.03 | 16.80 | 21.31 | 27.87 | 37.30 | 41.39 | 50.41 |
| | Kimina-Prover-RL-1.7B | 6.15 | 12.30 | 17.62 | 22.13 | 27.46 | 31.56 | 26.23 | 42.21 | 60.66 | 74.18 | 88.11 | 93.85 |
| | Kimina-Prover-Distill-8B | 7.38 | 11.89 | 16.39 | 24.18 | 28.69 | 32.38 | 24.59 | 44.26 | 61.89 | 75.00 | 85.25 | 89.34 |
| | Kimina-Prover-72B | 10.25 | 13.93 | 18.85 | 25.00 | 31.35 | 37.30 | 30.74 | 45.49 | 62.70 | 77.87 | 86.89 | 91.39 |
| *SoTA Tools* (text-based retrieval few-shot) | Kimina-Prover-RL-1.7B | 6.15 | 12.70 | 18.85 | 22.54 | 28.28 | 32.78 | 26.63 | 43.39 | 58.68 | 70.66 | 86.36 | 89.75 |
| | Kimina-Prover-Distill-8B | 8.61 | 13.52 | 21.72 | 28.28 | 34.02 | 36.07 | 28.28 | 46.28 | 59.92 | 71.90 | 83.47 | 86.07 |
| | Kimina-Prover-72B | 12.29 | 14.34 | 24.59 | 29.51 | 37.70 | 44.26 | 31.15 | 45.08 | **64.88** | 75.62 | 86.78 | 88.93 |
| *SoTA Two-Step* | Herald-Translator → Kimina-Prover-Distill-8B | 14.75 | 19.26 | 27.05 | 33.20 | 38.52 | 43.44 | 30.33 | 32.79 | 43.03 | 48.36 | 54.10 | 59.43 |
| *Our Tool* | **PROOFBRIDGE** (SFT only) | 6.97 | 13.52 | 19.26 | 24.18 | 29.92 | 34.84 | 27.87 | 45.90 | 60.66 | 66.39 | 72.13 | 78.69 |
| | **PROOFBRIDGE** (Retrieval-augmt. SFT) | 13.11 | 20.90 | 27.87 | 35.66 | 47.95 | 55.33 | 29.92 | 46.31 | 60.25 | 71.31 | 83.20 | 89.75 |
| | **PROOFBRIDGE** (Retrieval-augmt. SFT + Repair) | **16.39** | **25.41** | **29.51** | **37.70** | **50.41** | **62.70** | **32.79** | **47.13** | 64.75 | **78.69** | **90.16** | **95.49** |

## 5.6 ABLATION STUDIES

To assess the effect of in-context examples, Table 2 reports zero-shot, random few-shot, and text-based retrieval few-shot performance for three Kimina-Prover variants (72B, Distill-8B, and RL-1.7B). From the pass@32 results, Kimina-Prover-RL-1.7B achieves 40.16% SC and 75.00% TC in the zero-shot setting. When random examples are added, SC drops to 31.56% while TC rises to 93.85%, with similar trends across variants. This indicates that random examples improve TC (syntax) but hurt SC by causing the model to hallucinate semantically misaligned proofs. With text-based retrieval via Qwen3-Embedding-8B, SC rises to 32.38% but TC declines, likely because QA-based retrieval favors proofs with similar tactics, reducing tactic diversity. This highlights the need for retrieving semantically relevant examples via a DAG-aware encoder, as in PROOFBRIDGE.

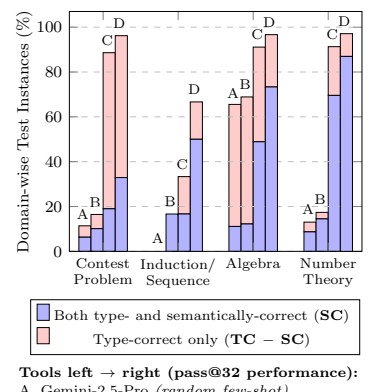

Tools left → right (pass@32 performance):
A. Gemini-2.5-Pro *(random few-shot)*
B. DeepSeek-Prover-V1.5-RL *(random few-shot)*
C. Kimina-Prover-72B *(text-based retr. few-shot)*
D. PROOFBRIDGE *(Retr-augmt. SFT + Repair)*

Figure 3: **Category-wise Results.** Proof auto-formalization performance across mathematical domains.

To quantify the contribution of each component of PROOF-BRIDGE, we perform an ablation over three variants. **PROOFBRIDGE (SFT)**, fine-tuned on labeled NL-FL pairs and evaluated in the few-shot setting with semantically relevant examples via our joint-embedding model, improves SC by +2.06% but reduces TC by −11.06% relative to Kimina-Prover-RL-1.7B (text-based retrieval few-shot). Next, in **PROOFBRIDGE (Retrieval-augmented SFT)**, we fine-tune the LLM with semantically relevant FL proofs included in the input, achieving +22.55% SC. **PROOFBRIDGE (Retrieval-augmented SFT + Repair)**, adding iterative proof repair, yields the best results: +29.92% SC and +5.74% TC over the same baseline. Relative to Kimina-Prover-RL-1.7B (random few-shot), the improvements are +31.14% SC and +1.64% TC.

## 6 CONCLUSION

We present PROOFBRIDGE, a unified framework for NL-to-Lean proof auto-formalization that translates both theorems and proofs end-to-end. At its core is a joint embedding model of NL and FL that encodes Lean proof DAGs, capturing tactic sequences and dependency structures. It enables highly effective cross-modal retrieval of semantically relevant FL proofs. These retrieved proofs act as demonstrations, guiding retrieval-augmented fine-tuning of an LLM. An iterative verifier-guided repair loop further refines generated proofs by combining Lean type-checking with semantic equivalence checking to ensure correctness. Evaluated on MINIF2F-TEST-PF, PROOFBRIDGE significantly outperforms state-of-the-art LLMs in both semantic correctness (by bi-directional equivalence proving) and type correctness, demonstrating that integrating structured embeddings, retrieval guidance, and verifier feedback leads to more reliable proof auto-formalization.

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

# A APPENDIX

## A.1 THE USE OF LARGE LANGUAGE MODELS (LLMS)

LLMs did *not* play a significant role in either the research ideation or the writing of this paper. Their use was limited to correcting minor grammatical issues and typographical errors.

## A.2 REPRODUCIBILITY AND DATASETS

All implementations in this work, including dataset construction, model training, cross-modal retrieval, inference, and evaluation, use Python 3.12.10 and Lean v4.15.0. Experiments were conducted on a high-performance AlmaLinux 9.5 (Teal Serval) cluster with a single Intel Xeon Platinum 8480+ CPU (32 cores, 2.0-4.0 GHz), 251 GiB of RAM, and one NVIDIA H100 GPU. Our full codebase, including scripts for dataset generation, model fine-tuning, and inference across both GPU- and API-based setups, is available at `https://github.com/PrithwishJana/ProofBridge`. The repository is well documented and includes instructions for creating a Python virtual environment, along with a comprehensive README detailing library dependencies, dataset formats, and step-by-step instructions to replicate the entire data pipeline and experimental workflow.

Through our repository, we release NUMINAMATH-LEAN-PF and MINIF2F-TEST-PF, following the respective licenses (Apache License 2.0 or MIT License) of their source datasets. Note that our NUMINAMATH-LEAN-PF dataset is carefully refined and extended from NuminaMath-LEAN (Wang et al., 2025), while MINIF2F-TEST-PF is curated by combining two versions of miniF2F-test (Zheng et al., 2022) from NuminaMath (2025) and Yang (2025). The original datasets were built by collecting problems from web sources, forums, and online documents, including worldwide competitions such as AMC, AIME, and IMO, as well as public K–12 exam papers. For each problem, the corresponding Lean theorem and proof were produced either by human annotators or via an LLM-based auto-formalizer. Formalizing large-scale proofs often requires an agile methodology (Jana & Pal, 2020; Jana et al., 2025), dividing proofs into small, manageable lemmas that can be handled by different human annotators. Many large-scale Lean projects now use tools such as `leanblueprint` (Massot, 2020), which serve as planning and coordination resources, enabling contributors to track progress, visualize dependencies, and identify remaining formalization tasks. Because the pipeline aggregates material from problem authors, human formalizers, LLM-based auto-formalizers, textbooks, competition archives, and exams, it is important to ensure proper privacy, licensing, and copyright compliance (Jana & Bandyopadhyay, 2013).

## A.3 TRAINING AND INFERENCE HYPERPARAMETERS

**NL/FL Cross-Modal Retrieval.** We train two dense encoders to embed NL and FL theorem+proof pairs into a shared semantic space. The NL encoder is initialized from `all-MiniLM-L6-v2` (Reimers & Gurevych, 2019), while the FL encoder builds on Lean-Dojo's (Yang et al., 2023) `ByT5`-based proof-state encoder, extended to process a linearized traversal of the proof DAG. Each encoder is equipped with a projection head that maps representations into a shared embedding space of dimension $d = 512$, with a dropout rate of $0.1$. During fine-tuning, we update only the top layers of each encoder to retain their pretrained linguistic and structural priors. Specifically, we train the last 3 layers of the NL encoder and the last 2 layers of the FL encoder. We set the maximum token length to 512 for both NL and FL sequences. The model is optimized using our symmetric contrastive objective (Equation 1) with temperature $\tau = 0.07$, trained using AdamW with a learning rate of $1 \times 10^{-5}$, weight decay of $0.01$, and a batch size of 32. Training is run for 10 epochs with gradient accumulation steps set to $4$. We also enable gradient checkpointing to reduce memory usage during fine-tuning.

**Proof Auto-Formalization.** PROOFBRIDGE builds on Kimina-Prover-RL-1.7B (Wang et al., 2025), which we further fine-tune for NL→FL translation using paired data $(M_{\text{NL}}, M_{\text{FL}})$ from NUMINAMATH-LEAN-PF. During both SFT and inference, our NL/FL cross-modal retrieval model gets the top-5 most relevant FL proofs from $\mathcal{D}$, which are provided as in-context demonstrations to guide Lean proof synthesis. We use the HuggingFace Trainer for supervised fine-tuning with the following settings: a per-device batch size of 8 and BF16 training enabled. Training is run for 5 epochs with a learning rate of $5 \times 10^{-6}$, cosine decay scheduling, and a warmup ratio of $0.05$. For all SoTA baselines in Table 2, we compute pass@k using stochastic decoding. Specifically, we run

LLM inference with a temperature of $0.6$ and top-p sampling of $0.95$, ensuring sufficient diversity across generated candidates.

## A.4 SEMANTIC EQUIVALENCE OF LEAN THEOREMS

Lean (Moura & Ullrich, 2021) is a functional programming language in which functions are first-class values, treated like any other data type (Jana, 2014). The task of auto-formalization is to convert a mathematical theorem and proof from natural language (NL) into a formal language (FL), such as Lean. When evaluating the performance of such systems, we propose two criteria for evaluation: *type correctness* and *semantic correctness*. Type correctness, which requires that the generated Lean proof is accepted by the Lean type-checker, is straightforward to verify and serves as the standard evaluation metric in the field. In contrast, semantic correctness, which ensures the FL theorem faithfully represents the meaning of the original NL theorem, presents a far greater challenge. To the best of our knowledge, such semantic equivalence has not been systematically evaluated in prior work. This section introduces a novel methodology to address this gap.

While directly measuring the semantic alignment between a NL theorem and a Lean theorem is an unsolved challenge, showing the logical equivalence of two Lean theorems is a tractable task. Our training dataset, NUMINAMATH-LEAN-PF, contains pairs of $\langle T_{\mathrm{NL}}, T_{\mathrm{FL}} \rangle$ where most of the $T_{\mathrm{FL}}$ were manually created by experts at Numina. We treat these high-quality $T_{\mathrm{FL}}$ theorems as gold-standard references, assuming they are faithful translations of their $T_{\mathrm{NL}}$ counterparts. This allows us to reduce the intractable problem of verifying a model's generated theorem $\widetilde{T}_{\mathrm{FL}}$ against the original $T_{\mathrm{NL}}$ to the more tractable task of checking for logical equivalence between $\widetilde{T}_{\mathrm{FL}}$ and the gold-standard reference $T_{\mathrm{FL}}$, which can be checked in Lean itself.

To be more specific, we enforce this semantic equivalence check by proving the logical biconditional $\widetilde{T}_{\mathrm{FL}} \leftrightarrow T_{\mathrm{FL}}$ in Lean. Theorems like $\widetilde{T}_{\mathrm{FL}}$ and $T_{\mathrm{FL}}$ are of type `Prop` in Lean. The following theorem from Mathlib states that for any two propositions, a logical biconditional between two propositions is itself logically equivalent to their propositional equality:

```
theorem propext_iff{a b : Prop} :
a = b ↔ (a ↔ b)
```

The task thus converts to proving the equality $\widetilde{T}_{\mathrm{FL}} = T_{\mathrm{FL}}$ within Lean. This requires clarifying the specific notion of equality being used, as Lean distinguishes between three primary types: *syntactic*, *definitional*, and *propositional* (Buzzard, 2022). Syntactic equality is the strictest form of equality in Lean, as it only admits expressions that are structurally identical according to their Abstract Syntax Trees, without any computation or reduction. Definitional equality is a more relaxed form of equality than syntactic equality, where two expressions are considered equal if they compute or reduce to the same normal form. Propositional equality is the weakest form of equality, and also the standard notion of equality used in mathematical theorems. Two terms `a, b` are propositionally equal in Lean if we can construct a proof term for the proposition `a = b`.

For our evaluation, we seek to measure how closely a $\widetilde{T}_{\mathrm{FL}}$ matches the $T_{\mathrm{FL}}$. The strictest criterion, syntactic equality, is too restrictive given the current state-of-the-art, as it would fail valid theorems with trivial notational differences. Conversely, full propositional equality can be too permissive; a proof of equivalence can be arbitrarily complex, making it difficult to automate and decide.

Therefore, we adopt a pragmatic compromise: we check for **definitional equality** supplemented by a form of **bounded propositional equality**. This means we primarily check if $\widetilde{T}_{\mathrm{FL}}$ and $T_{\mathrm{FL}}$ reduce to the same normal form, but we also permit some propositional equality, provided they can be proven using a collection of tactics so that their proof complexity is bounded.

We then leverage Gemini 2.5 Pro as an automated equivalence checker. The model is prompted to synthesize a proof for the biconditional theorem ($\widetilde{T}_{\mathrm{FL}} \leftrightarrow T_{\mathrm{FL}}$), with instructions limiting it to a specific subset of available tactics. This restricted set includes three powerful automated tactics, `rfl`, `simp`, and `ring`, each is designed to discharge a specific class of goals: `rfl` for definitional equality, `simp` for simplification, and `ring` for polynomial identities.

As definitional equality is our primary target, the equivalence checker first attempts to solve the goal with the `rfl` tactic. This single tactic should suffice for the majority of cases. If `rfl` fails, the

checker then tries `simp`. This tactic performs additional simplifications by rewriting the goal using theorems from Mathlib that are tagged for its use. Critically, we use `simp` without any arguments. Providing explicit arguments would require a demanding search for the correct lemmas and could introduce unbounded complexity, violating our goal of a bounded proof search. Furthermore, the need for `simp` with arguments could imply that the required rewrite is non-trivial, since the default simplification set contains most of the trivial facts [1]. Since our goal is to ensure a close correspondence between $\widetilde{T}_{\text{FL}}$ and $T_{\text{FL}}$, a proof requiring such a targeted rewrite indicates a semantic distance that we classify as a mismatch. The `ring` tactic is a valuable complement to the previous tactics as it specializes in proving polynomial equalities. The `ring` tactic operates by reducing arithmetic expressions to a canonical normal form. This allows it to prove the equivalence of expressions that are algebraically identical but not definitionally so, such as `x^2` and `x * x`, which `rfl` and default `simp` would otherwise fail to solve.

The three tactics discussed above cover most of the direct equivalences we aim to check. The remaining tactics in our instruction set are designed for a more nuanced case: proving the biconditional when two theorems differ only in their use of auxiliary variables. We observed that human experts and language models may make different but equally valid decisions on whether to introduce an auxiliary variable. We therefore classify such theorems as equivalent. For example, consider the following:

```
def Prop1 := (∀ (b h v : ℝ), (0 < b ∧ 0 < h ∧ 0 < v) → (v = 1 / 3 * (b *
    h)) → (b = 30) → (h = 13 / 2) → v = 65)
def Prop2 := (∀ {B h : ℝ}, (B = 30) → (h = 6.5) → (1 / 3) * B * h = 65)
example : Prop1 ↔ Prop2 := by
  constructor
  · intro
    simp
    ring
  · simp
    intros
    nlinarith
```

The main difference between the two propositions `Prop1` and `Prop2` is the presence of the auxiliary variable `v` in one. To prove that such theorems are equivalent, one must typically prove the biconditional by separately proving the implications of both direction. This requires a step-by-step proof construction, and the tactics above are included in our instruction set.

Finally, we note that the LLM judge's role is only to synthesize a biconditional proof under the bounded tactic set described above; the produced proof is then fully type-checked by the Lean kernel, so the SC decision ultimately depends on Lean's verifier (making the metric conservative rather than prone to false positives).

**Comparison with Existing Semantic Correctness Metrics.** Prior work has proposed similar semantic correctness metrics, including BEq (Liu et al., 2025; Wu et al., 2026) and its extensions BEq+ (Poiroux et al., 2025). While the general idea behind our SC metric and BEq is similar, both aiming to establish bidirectional equivalence in Lean, the sets of allowed tactics differ. Because the notion of equivalence depends on the permitted tactics, these differences lead to meaningful distinctions between SC and BEq. BEq+ is a reference-based metric inspired by BEq and uses a set of tactics comparable to SC. However, BEq+ is deterministic and CPU-efficient, while SC relies on an LLM-based proof synthesizer. This creates a trade-off: the LLM can capture equivalences beyond the reach of the deterministic procedure, whereas BEq+ provides a reproducible evaluation.

Consider `Prop1` and `Prop2` above as an example. `Prop1` (gold-standard FL theorem from miniF2F-Test-PF) explicitly introduces an auxiliary variable `v` to denote volume, whereas `Prop2` (produced by Kimina-Prover-RL-1.7B) omits the auxiliary variable and substitutes the corresponding formula directly. The tactic set allowed by BEq is not expressive enough to establish equivalence in such cases, so these theorems would not be recognized as equivalent under BEq. Our SC metric, by contrast, was specifically designed to handle such variations, reflecting the fact that human experts may also differ in whether they introduce auxiliary variables. By explicitly handling these variations

---

[1]It is important to note that the default simp set intentionally excludes lemmas like associativity and commutativity, as they can cause the simplifier to loop indefinitely. However, since these lemmas primarily concern algebraic expressions, they can be handled by the `ring` tactic.

and using LLM-generated bidirectional proofs that are type-checked, SC provides an evaluation that is both more lenient and faithful in assessing the performance of auto-formalization models.

## A.5    ILLUSTRATIVE EXAMPLE

We present an example of an NL theorem+proof pair from MINIF2F-TEST-PF and compare the `pass@1` output of proof auto-formalization generated by Kimina-Prover-RL-1.7B in the text-based retrieval few-shot setting with that produced by PROOFBRIDGE (using Retrieval-augmented SFT + Repair). We first show the retrievals of semantically relevant FL theorem+proof pairs from $\mathcal{D}$, followed by the `pass@1` output proof auto-formalization generated by PROOFBRIDGE. In this example, the output by Kimina-Prover-RL-1.7B is type-correct (TC) but not semantically correct (SC), i.e., it is not bi-directionally equivalent to the gold-standard Lean proof. In contrast, the output by PROOFBRIDGE is both TC and SC.

---

**Input NL theorem+proof pair** (MINIF2F-TEST-PF)

`<informal_theorem>`
A point $(x, y)$ on the coordinate plane with both coordinates negative is a distance of 6 units from the $x$-axis. It is a distance of 15 units from the point $(8, 3)$. It is a distance $\sqrt{n}$ from the origin. What is $n$? Show that it is 52.
`</informal_theorem>`

`<informal_proof>`
We know that $y = -6$ from the given information. By the distance formula, we have the equation $\sqrt{(x-8)^2 + (-6-3)^2} = 15$. Solving, we have

$$\sqrt{(x-8)^2 + (-6-3)^2} = 15$$
$$x^2 - 16x + 64 + 81 = 225$$
$$x^2 - 16x - 80 = 0$$
$$(x-20)(x+4) = 0$$

Thus, $x + 4 = 0$ or $x - 20 = 0$, so $x = -4$ or $x = 20$. $x = -4$ by the given conditions. Thus, our point is $(-4, -6)$ and is a distance of $\sqrt{(-4)^2 + (-6)^2} = \sqrt{52}$ units from the origin. $n = 52$.
`</informal_proof>`

---

**Gold-standard Lean theorem** (MINIF2F-TEST-PF)

```
import Mathlib
import Aesop
set_option maxHeartbeats 0
open BigOperators Real Nat Topology Rat

theorem mathd_algebra_288 (x y : ℝ) (n : NNReal) (h₀ : x < 0 ∧ y < 0) (h₁ : abs y = 6)
    (h₂ : Real.sqrt ((x - 8) ^ 2 + (y - 3) ^ 2) = 15)
    (h₃ : Real.sqrt (x ^ 2 + y ^ 2) = Real.sqrt n) : n = 52 := by
    sorry
```

---

**Output Lean theorem+proof pair by Kimina-Prover-RL-1.7B**
(text-based retrieval few-shot setting)

```
import Mathlib

/- A point (x,y) on the coordinate plane with both coordinates negative is a distance of
    6 units from the x-axis. It is a distance of 15 units from the point (8,3). It is a
    distance \sqrt{n} from the origin. What is n? Show that it is 52.-/

theorem algebra_60927 (n : ℕ) (hn : n ≥ 1) (h : ∃ x y : ℝ, x < 0 ∧ y < 0 ∧ |y| = 6 ∧ √
    ((x - 8)^2 + (y - 3)^2) = 15 ∧ √(x^2 + y^2) = √n) : n = 52 := by
  rcases h with ⟨x, y, hx, hy, h1, h2, h3⟩
  have hy1 : y = -6 := by
    cases' abs_cases y with hy2 hy2
    · linarith
    · linarith
  have h2' : (x - 8)^2 + (y - 3)^2 = 225 := by
    calc
```

```
     (x - 8)^2 + (y - 3)^2 = (√((x - 8)^2 + (y - 3)^2)) ^ 2 := by rw [Real.sq_sqrt];
   positivity
     _ = 15 ^ 2 := by rw [h2]
     _ = (225 : ℝ) := by norm_num
 rw [hy1] at h2'
 have h2x : (x - 8)^2 = 144 := by nlinarith
 have hx1 : x = -4 := by
   nlinarith [h2x, hx]
 have h3' : x^2 + y^2 = (n : ℝ) := by
   calc
     x^2 + y^2 = (√(x^2 + y^2)) ^ 2 := by rw [Real.sq_sqrt]; positivity
     _ = (√n) ^ 2 := by rw [h3]
     _ = (n : ℝ) := by rw [Real.sq_sqrt]; positivity
 rw [hx1, hy1] at h3'
 have hn1 : (n : ℝ) = 52 := by nlinarith
 have hn2 : n = 52 := by
   exact_mod_cast hn1
 exact_mod_cast hn2
```

Note that, although the Lean theorem generated by Kimina-Prover-RL-1.7B is type-correct (TC), it differs semantically from the gold-standard theorem. The main difference lies in the quantification of variables. In the gold-standard theorem, the variables $x, y, n$ are universally quantified as explicit arguments, and all hypotheses are stated as direct assumptions; this asserts that for any triple $(x, y, n)$ satisfying the geometric constraints, $n = 52$. In contrast, the Kimina-Prover-RL-1.7B output universally quantifies $n$ but existentially quantifies $x$ and $y$ within the hypotheses. This more accurately reflects the intended geometric meaning: for a given $n$ satisfying the distance constraints, there exists a point $(x, y)$ realizing those constraints, and consequently $n = 52$. Therefore, while both theorems are syntactically valid in Lean, they encode slightly different logical statements. This difference prevents an LLM judge from producing a type-checkable proof of bi-directional equivalence between the two theorems. As a result, the Kimina-Prover-RL-1.7B's output is not semantically correct (SC).

---

**Comparison between the gold-standard theorem and Kimina-Prover-RL-1.7B's output**

```
/- Gold-standard theorem -/
theorem mathd_algebra_288 (x y : ℝ) (n : NNReal) (h₀ : x < 0 ∧ y < 0) (h₁ : abs y = 6)
    (h₂ : Real.sqrt ((x - 8) ^ 2 + (y - 3) ^ 2) = 15)
    (h₃ : Real.sqrt (x ^ 2 + y ^ 2) = Real.sqrt n) : n = 52 := by
    sorry

/- Kimina-Prover-RL-1.7B output theorem -/
theorem algebra_60927 (n : ℕ) (hn : n ≥ 1) (h : ∃ x y : ℝ, x < 0 ∧ y < 0 ∧ |y| = 6 ∧ √
    ((x - 8)^2 + (y - 3)^2) = 15 ∧ √(x^2 + y^2) = √n) : n = 52 := by
    sorry
```

---

Below, we present the relevant FL theorem+proof pairs (demonstrations) from $\mathcal{D}$ retrieved by PROOFBRIDGE, along with their relevance scores.

---

**Lean theorem+proof pairs retrieved by PROOFBRIDGE's NL/FL Cross-Modal Retrieval**

### Relevant Lean theorem+proof 1, with relevance score 0.786764 out of 1.0

```
import Mathlib

/- Find the distance between the points $(2,2)$ and $(-1,-1)$. -/
theorem algebra_13734 (p1 p2 : ℝ × ℝ) (hp1 : p1 = (2, 2)) (hp2 : p2 = (-1, -1)) :
    Real.sqrt ((p1.1 - p2.1)^2 + (p1.2 - p2.2)^2) = 3 * Real.sqrt 2 := by
  rw [hp1, hp2]
  norm_num
  ring
  rw [Real.sqrt_eq_iff_sq_eq] <;> norm_num
  ring
  norm_num
```

### Relevant Lean theorem+proof 2, with relevance score 0.768226 out of 1.0

```
import Mathlib
open Real

/- Prove that the angle (in degrees) between the vectors $(2,5)$ and $(-3,7)$ is $45$. -/
theorem calculus_17161 :
    arccos ((2 * (-3) + 5 * 7) / (sqrt (2 ^ 2 + 5 ^ 2) * sqrt ((-3) ^ 2 + 7 ^ 2))) * 180
    / π = 45 := by
```

```
  have h1 : (2 * (-3) + 5 * 7 : ℝ) / (sqrt (2 ^ 2 + 5 ^ 2) * sqrt ((-3) ^ 2 + 7 ^ 2)) =
    Real.sqrt 2 / 2 := by
    have h2 : sqrt ((2 : ℝ) ^ 2 + (5 : ℝ) ^ 2) = Real.sqrt 29 := by
      norm_num [Real.sqrt_eq_iff_sq_eq]

    have h3 : sqrt ((-3 : ℝ) ^ 2 + (7 : ℝ) ^ 2) = Real.sqrt 58 := by
      norm_num [Real.sqrt_eq_iff_sq_eq]

    have h4 : (2 * (-3) + 5 * 7 : ℝ) = 29 := by norm_num

    rw [h2, h3, h4]

    have h5 : Real.sqrt 29 * Real.sqrt 58 = Real.sqrt 2 * (29 : ℝ) := by
      calc
        Real.sqrt 29 * Real.sqrt 58 = Real.sqrt (29 * 58 : ℝ) := by
          rw [← Real.sqrt_mul (by norm_num)]
        _ = Real.sqrt ((2 : ℝ) * (29 ^ 2 : ℝ)) := by norm_num
        _ = Real.sqrt (2 : ℝ) * Real.sqrt ((29 : ℝ) ^ 2 : ℝ) := by
          rw [Real.sqrt_mul (by norm_num)]
        _ = Real.sqrt (2 : ℝ) * (29 : ℝ) := by
          rw [Real.sqrt_sq (by norm_num)]

    field_simp [h5]
    <;> ring_nf <;> norm_num [Real.sq_sqrt]

  rw [h1]

  have h5 : arccos (Real.sqrt 2 / 2) = Real.pi / 4 := by
    have h6 : Real.sqrt 2 / 2 = Real.cos (Real.pi / 4) := by
      rw [Real.cos_pi_div_four]
      <;> ring_nf <;> norm_num
      <;> ring
    rw [h6]
    have h7 : arccos (Real.cos (Real.pi / 4)) = Real.pi / 4 := by
      apply arccos_cos
      all_goals linarith [Real.pi_pos]
    exact h7

  rw [h5]

  field_simp [Real.pi_pos]
  <;> linarith [Real.pi_gt_three]
```

### Relevant Lean theorem+proof 3, with relevance score 0.765285 out of 1.0

```
import Mathlib

/- Show that the sum of $\sqrt{3x^2 + 2x + 1}$ and $\sqrt{3x^2 - 4x + 2}$ is at least $
    \sqrt{51}/3$ for all real $x$. -/
theorem inequalities_201318 (x : ℝ) :
    Real.sqrt (3 * x^2 + 2 * x + 1) + Real.sqrt (3 * x^2 - 4 * x + 2) ≥
    Real.sqrt 51 / 3 := by
  set y := Real.sqrt (3 * x^2 + 2 * x + 1)
  set z := Real.sqrt (3 * x^2 - 4 * x + 2)
  have hy2 : y^2 = 3 * x^2 + 2 * x + 1 := by
    rw [Real.sq_sqrt]
    nlinarith [sq_nonneg (x + 1 / 3)]
  have hz2 : z^2 = 3 * x^2 - 4 * x + 2 := by
    rw [Real.sq_sqrt]
    nlinarith [sq_nonneg (x - 2 / 3)]
  have hy4_pos : 0 ≤ (3 * x^2 + 2 * x + 1 : ℝ) := by
    nlinarith [sq_nonneg (x * 3 + 1)]
  have hz4_pos : 0 ≤ (3 * x^2 - 4 * x + 2 : ℝ) := by
    nlinarith [sq_nonneg (x * 3 - 2)]
  have h11 : (Real.sqrt 51 / 3) ^ 2 = (51 / 9 : ℝ) := by
    calc
      (Real.sqrt 51 / 3) ^ 2 = (Real.sqrt 51) ^ 2 / 9 := by ring
      _ = (51 / 9 : ℝ) := by
        rw [Real.sq_sqrt (by norm_num)]
        <;> ring
  have h50 : (y + z) ^ 2 ≥ (Real.sqrt 51 / 3) ^ 2 := by
    nlinarith [sq_nonneg (y - z), sq_nonneg (x - 2 / 3), sq_nonneg (x + 1 / 3),
      h11, Real.sqrt_nonneg 51, Real.sq_sqrt (show 0 ≤ (51 : ℝ) by norm_num),
      mul_nonneg (Real.sqrt_nonneg (3 * x^2 + 2 * x + 1)) (Real.sqrt_nonneg (3 * x^2 - 4 *
      x + 2)),
      sq_nonneg (y ^ 2 - z ^ 2), sq_nonneg (y * z - Real.sqrt ((3 * x^2 + 2 * x + 1) * (3
      * x^2 - 4 * x + 2)))
      ]
```

```
  have h51 : (y + z) ≥ 0 := by positivity
  have h54 : (Real.sqrt 51 / 3) ≥ 0 := by positivity
  have h52 : (y + z) ≥ (Real.sqrt 51 / 3) := by
    have h15 : (y + z) ^ 2 ≥ (Real.sqrt 51 / 3) ^ 2 := h50
    have h16 : (Real.sqrt 51 / 3) ≥ 0 := h54
    have h17 : (y + z) - (Real.sqrt 51 / 3) ≥ 0 := by
      nlinarith [sq_nonneg ((y + z) - (Real.sqrt 51 / 3)),
        sq_nonneg ((y + z) + (Real.sqrt 51 / 3)),
        Real.sqrt_pos.mpr (show (0 : ℝ) < 51 by linarith : (51 : ℝ) > 0)
        ]
    linarith
  linarith
```

### Relevant Lean theorem+proof 4, with relevance score 0.764932 out of 1.0

```
import Mathlib
open Real Set
open scoped BigOperators

/- Given that $\sqrt{(x-3)^2 + (y+4)^2} + \sqrt{(x+5)^2 + (y-8)^2} = 20$, prove that the
    distance between $(-5,8)$ and $(3,-4)$ is $4\sqrt{13}$. -/
theorem algebra_10804 {x y : ℝ} (h : sqrt ((x - 3)^2 + (y + 4)^2) + sqrt ((x + 5)^2 + (y
    - 8)^2) = 20) :
    sqrt ((-5 - 3)^2 + (8 - (-4))^2) = 4 * sqrt 13 := by
  have h1 : sqrt ((-5 - 3)^2 + (8 - (-4))^2) = sqrt 208 := by
    norm_num
  rw [h1]
  have h2 : sqrt 208 = 4 * sqrt 13 := by
    rw [Real.sqrt_eq_iff_sq_eq] <;> norm_num
    <;> ring_nf <;> norm_num
  rw [h2]
  all_goals norm_num
```

### Relevant Lean theorem+proof 5, with relevance score 0.755091 out of 1.0

```
import Mathlib

/- Show that the square root of $(-3)^2$ is $3$, that $3$ is non-negative, and that $3$
    squared equals $(-3)$ squared. -/
theorem algebra_8272 : Real.sqrt ((-3)^2) = 3 ∧ 0 ≤ 3 ∧ 3^2 = (-3)^2 := by
  have h1 : Real.sqrt ((-3 : ℝ) ^ 2) = 3 := by
    rw [Real.sqrt_sq_eq_abs]
    norm_num
  constructor
  · exact h1
  constructor
  · norm_num
  · norm_num
```

The proof auto-formalization generated by PROOFBRIDGE is as follows:

---

**Output Lean theorem+proof pair by PROOFBRIDGE** (Retrieval-augmented SFT + Repair)

```
import Mathlib

/- A point (x,y) on the coordinate plane with both coordinates negative is a distance of
    6 units from the x-axis. It is a distance of 15 units from the point (8,3). It is a
    distance \sqrt{n} from the origin. What is n? Show that it is 52.-/

theorem algebra_53819 (x y : ℝ) (hx : x < 0) (hy : y < 0) (h1 : abs y = 6) (h2 :
    Real.sqrt ((x - 8) ^ 2 + (y - 3) ^ 2) = 15) :
    Real.sqrt (x ^ 2 + y ^ 2) = Real.sqrt 52 := by
  have hy6 : y = -6 := by
    cases eq_or_eq_neg_of_abs_eq h1 with
    | inl h =>
      linarith
    | inr h =>
      linarith
  have h3 : (x - 8) ^ 2 + (y - 3) ^ 2 = 225 := by
    calc
      (x - 8) ^ 2 + (y - 3) ^ 2 = (Real.sqrt ((x - 8) ^ 2 + (y - 3) ^ 2)) ^ 2 := by
        rw [Real.sq_sqrt]
        positivity
      _ = 15 ^ 2 := by
        rw [h2]
      _ = 225 := by
```

```
      norm_num
  rw [hy6] at h3
  have h4 : (x - 8) ^ 2 = 144 := by
    nlinarith
  have h5 : x - 8 = 12 ∨ x - 8 = -12 := by
    have h6 : (x - 8) ^ 2 - 144 = 0 := by linarith
    have h7 : (x - 8 - 12) * (x - 8 + 12) = 0 := by
      nlinarith
    cases (mul_eq_zero.mp h7) with
    | inl h8 =>
      left
      linarith
    | inr h9 =>
      right
      linarith
  cases h5 with
  | inl h6 =>
    have hx20 : x = 20 := by linarith
    linarith
  | inr h6 =>
    have hx4 : x = -4 := by linarith
    have h7 : x ^ 2 + y ^ 2 = 52 := by
      rw [hx4, hy6]
      norm_num
    have h8 : Real.sqrt (x ^ 2 + y ^ 2) = Real.sqrt 52 := by
      rw [h7]
    exact h8
```

The Lean theorem generated by PROOFBRIDGE and the gold-standard theorem are semantically equivalent. Both correctly capture the intended geometric scenario: the point with negative coordinates satisfying the given distance constraints is at a distance of $\sqrt{52}$ from the origin. Using our set of restricted tactics, the LLM judge is able to generate a type-checkable proof establishing the bi-directional equivalence between the two theorems. Therefore, the PROOFBRIDGE output is semantically correct (SC).

---

**Comparison between the gold-standard theorem and PROOFBRIDGE's output**

```
/- Gold-standard theorem -/
theorem mathd_algebra_288 (x y : ℝ) (n : NNReal) (h₀ : x < 0 ∧ y < 0) (h₁ : abs y = 6)
    (h₂ : Real.sqrt ((x - 8) ^ 2 + (y - 3) ^ 2) = 15)
    (h₃ : Real.sqrt (x ^ 2 + y ^ 2) = Real.sqrt n) : n = 52 := by
    sorry

/- ProofBridge output theorem -/
theorem algebra_53819 (x y : ℝ) (hx : x < 0) (hy : y < 0) (h1 : abs y = 6) (h2 :
    Real.sqrt ((x - 8) ^ 2 + (y - 3) ^ 2) = 15) :
    Real.sqrt (x ^ 2 + y ^ 2) = Real.sqrt 52 := by
  sorry
```

---

**Lean proof of bi-directional equivalence between PROOFBRIDGE's output theorem and the gold-standard**

```
import Mathlib.Analysis.SpecialFunctions.Sqrt
import Mathlib.Data.NNReal.Basic
open BigOperators Real Nat Topology Rat

-- We define the two statements as propositions to prove their equivalence.

def mathd_algebra_288_stmt :=
  ∀ (x y : ℝ) (n : NNReal) (h₀ : x < 0 ∧ y < 0) (h₁ : abs y = 6)
    (h₂ : Real.sqrt ((x - 8) ^ 2 + (y - 3) ^ 2) = 15)
    (h₃ : Real.sqrt (x ^ 2 + y ^ 2) = Real.sqrt n), n = 52

def algebra_53819_stmt :=
  ∀ (x y : ℝ) (hx : x < 0) (hy : y < 0) (h1 : abs y = 6)
    (h2 : Real.sqrt ((x - 8) ^ 2 + (y - 3) ^ 2) = 15),
    Real.sqrt (x ^ 2 + y ^ 2) = Real.sqrt 52

-- Proof of bidirectional equivalence
theorem equivalence : mathd_algebra_288_stmt ↔ algebra_53819_stmt := by
  constructor
  -- Proof that mathd_algebra_288_stmt implies algebra_53819_stmt
  · intro h_mathd_stmt
```

```
    intro x y hx hy h1 h2
    -- To use h_mathd_stmt, we need to provide a value for `n` and a proof for `h₃`.
    -- Let's define `n` as the value of 'x^2 + y^2'.
    -- First, we need to show 'x^2 + y^2' is non-negative to create an `NNReal`.
    have h_nonneg : 0 ≤ x ^ 2 + y ^ 2 := by positivity
    let n : NNReal := ⟨x ^ 2 + y ^ 2, h_nonneg⟩
    -- The hypothesis `h₃` now holds by definition of `n`.
    have h3 : Real.sqrt (x ^ 2 + y ^ 2) = Real.sqrt n := by
      simp [n, NNReal.coe_mk]
    -- The hypothesis `h₀` is equivalent to `hx` and `hy`.
    have h0 : x < 0 ∧ y < 0 := ⟨hx, hy⟩
    -- Now we can apply the first statement.
    have n_eq_52 : n = 52 := h_mathd_stmt x y n h0 h1 h2 h3
    -- Our goal is `Real.sqrt (x ^ 2 + y ^ 2) = Real.sqrt 52`.
    -- We can rewrite `h3` using `n_eq_52` to achieve the goal.
    rwa [n_eq_52] at h3

  -- Proof that algebra_53819_stmt implies mathd_algebra_288_stmt
  · intro h_algebra_stmt
    intro x y n h0 h1 h2 h3
    -- The hypotheses of the second statement are all available.
    have h_sqrt_val : Real.sqrt (x ^ 2 + y ^ 2) = Real.sqrt 52 :=
      h_algebra_stmt x y h0.left h0.right h1 h2
    -- We are given `h3`: `Real.sqrt (x ^ 2 + y ^ 2) = Real.sqrt n`.
    -- By transitivity, `Real.sqrt n = Real.sqrt 52`.
    have sqrt_n_eq_sqrt_52 : Real.sqrt n = Real.sqrt 52 := by
      rw [← h3, h_sqrt_val]
    -- Since `Real.sqrt` is injective on non-negative numbers, `n` must equal `52`.
    -- We get the equality on `ℝ` first.
    have n_val_eq_52 : (n : ℝ) = 52 :=
      (Real.sqrt_inj n.property (by norm_num)).mp sqrt_n_eq_sqrt_52
    -- Then we lift this equality to `NNReal`.
    exact NNReal.eq n_val_eq_52
```

