# OpenReview forum: "ProofBridge: Auto-Formalization of Natural Language Proofs in Lean via Joint Embeddings"
_ICLR.cc/2026/Conference — ICLR 2026 Poster_

### Official Review · Reviewer_B6B5 · 2025-10-27

**Soundness:** 3
**Presentation:** 4
**Contribution:** 3
**Rating:** 6
**Confidence:** 4

**Summary:**

ProofBridge is a unified framework for directly translating both the statement and proof from natural to formal language. It utilizes an embedding system that projects FL and NL statements to a joint space. This is used as a retriever for the autoformalization model which the authors train using SFT. The model is tested with iterative repair and it demonstrates superior performance in type check rate and semantic correctness compared to baselines.

**Strengths:**

The task of end-to-end proof autoformalization has, as the authors claim, been underexplored and it is an interesting new problem formulation. It is well-motivated by the difficulty manual formalization, such as during the AlphaProof experiment on IMO 2024 problems. Although there has been existing work on statement or proof autoformalization, an end-to-end approach for both statements and proofs has rarely been explored or evaluated. The experiment results rigorously show the proposed method's advantage over the numerous baselines considered. This paper may serve as a solid baseline for this end-to-end autoformalization setting for the community.

The presentation of this paper is clear and easy to follow.

**Weaknesses:**

My biggest concern is that while the problem setting has been underexplored, the method that the authors consider is largely only piecing together existing approaches. The method largely resembles existing work on retrieval-augmented statement autoformalization (see below), with the added complexity that it starts training from Kimina-Prover-1.7B which has been trained for proving already.

Contrary to what the authors claim, I think NL/FL joint embedding has been explored in previous work, in the form of first informalizing the FL statement and then embedding NL queries and informalized theorems into a joint space, and retrieving similar FL examples from NL queries at test time for the purpose of statement or proof autoformalization. This was proposed in prior work like [1, 2, 3]. In this paper, the authors proposed a slightly different method that trains linear projections from FL and NL embeddings (based on existing models) into a shared space. Although technically different, I don't see how this is advantageous compared to existing approaches based on informalization which have shown to be effective. Either some explanation or quantitative comparison is needed to demonstrate this.

[1] https://arxiv.org/abs/2403.13310v2
[2] https://arxiv.org/abs/2410.10878
[3] https://openreview.net/forum?id=hUb2At2DsQ

The "Semantic Correctness" (SC) evaluation metric is not new. In [1], the BEq metric was introduced which is basically the same as SC. Subsequent work such as RLMEval [2] and StepFun-Formalizer [3] have extended and improved BEq (although I admit the latter should be considered contemporaneous work). Even ignoring BEq, there has been extensive prior work on using an LLM judge as a semantic correctness check. This is a small issue, but I would encourage the authors to recognize the previous work.

[1] https://openreview.net/forum?id=hUb2At2DsQ
[2] https://arxiv.org/abs/2406.07222
[3] https://arxiv.org/abs/2508.04440

The retriever baseline, all-MiniLM-L6-v2, shouldn't be considered a SOTA. That is a small model pretrained on general NL data. I encourage the authors to try a larger pretrained model such as e5-mistral-7b-instruct (which LeanSearch seems to use), or at least not label the current baseline as SOTA (because LeanSearch exists).

**Questions:**

For the same natural language problem, there might exist different correct ways to state it in Lean, some being easier to prove than others. Do the authors envision ProofBridge to be extensible to optimize for downstream provability of formalized statements?

---

> ### Author Response · Authors · 2025-11-22
> **Authors' response to Reviewer B6B5 (Part 1/3)**
>
> We sincerely thank the reviewer for their thoughtful feedback and constructive suggestions. We also appreciate the valuable time spent preparing a detailed review. In response, we have revised the paper to include new comparative results against SoTA models, more detailed ablation studies, and a qualitative example. Below, we address the comments, organized under Weaknesses (W) and Questions (Q). We hope that these changes make the paper more thorough and comprehensive, and address the reviewer’s concerns.
>
> ------
>
> **_[W1+W2] …method is largely only piecing together existing approaches…resembles existing work on retrieval-augmented statement autoformalization…NL/FL joint embedding has been explored in previous work…don't see how this is advantageous compared to existing approaches based on informalization which have shown to be effective…explanation or quantitative comparison is needed to demonstrate this._**
>
> We thank the reviewer for pointing out these papers and raising this concern. We have now **cited and discussed** all three works in _Section 2 (Related Work)_ of the **revised submission**. The cited works focus only on FL theorem (statement) retrieval and do not retrieve proofs. Retrieving FL proofs is a different problem, and our empirical results show that encoding Lean proofs as plain text is not effective because many proofs share similar tactics and token patterns. In contrast, ProofBridge uses a DAG-aware FL encoder and a contrastive joint NL-FL embedding that captures proof structure, leading to significantly better separation in embedding space and higher Recall and MRR for NL-to-FL theorem+proof retrieval, which enables effective retrieval-augmented SFT for proof auto-formalization. Below, we clarify how our problem setting and methodology differ from the cited work:
>
> - **Comparison with the FL theorem retrieval mechanism in “A Semantic Search Engine for Mathlib4”[1].** This system informalizes FL statements from mathlib and indexes them for semantic search. The task is NL query → FL theorem retrieval; the FL proofs are neither stored nor retrieved. The system also relies on an off-the-shelf plain-text encoder (E5-Mistral-7B-Instruct). In contrast, ProofBridge’s problem setting is NL theorem+proof → FL theorem+proof retrieval, where retrieving the proof is essential. We argue below that off-the-shelf plain-text encoders are ineffective for retrieving FL proofs:
>
>     * **Empirical comparison (_Section 5.5, Table 1_).** In the **revised submission**, we compare ProofBridge’s joint-embedding model against encoding NL and FL theorem-proof pairs using three raw-text encoders:
>
>         **(a)** all-MiniLM-L6-v2 (Baseline Encoder, 22.7M parameters)
>
>         **(b)** Qwen3-Embedding-8B (SoTA Encoder, as suggested by Reviewer 9ttn)
>
>         **(c)** E5-Mistral-7B-Instruct (SoTA Encoder, as suggested by Reviewer B6B5)
>
>         We also introduce a **new metric**, mean Median Gap (mMG), which quantifies how well the retrieved and non-retrieved instances are separated in the embedding space (_detailed in Section 5.2_).
>
>         As shown in _Section 5.5_, ProofBridge outperforms both the Baseline and SoTA encoders in Recall, MRR, and mMG, despite being roughly 32x smaller than the SoTA.
>
>     * **Why SoTA text plain-text encoders underperform? (_detailed in Section 5.5 of the revised submission_)**
>
>         **(a)** **QA-oriented, text-based encoders capture only coarse domain signals.** These models are trained on broad corpora (e.g., legal, medical), causing many mathematical texts to collapse into similar regions of the embedding space. In Table 2, for E5-Mistral-7B-Instruct and Qwen3-Embedding-8B in NL-to-FL retrieval, the median cosine similarity of top-K retrieved examples is very close to that of non-retrieved ones (mMG = 0.10 and 0.29, respectively). In contrast, ProofBridge achieves a significantly larger separation (mMG = 0.65).
>
>         **(b)** **SoTA text encoders depend heavily on lexical similarity, ineffective for Lean proofs.** Raw-text encoders often rely on superficial lexical cues such as keywords. In Lean, many proofs use the same tactics and similar surface tokens, so keyword overlap becomes non-discriminative. ProofBridge instead incorporates the DAG structure of proofs and learns a contrastive joint alignment using a DAG-aware FL encoder that captures fine-grained proof semantics. For NL-to-FL retrieval, ProofBridge achieves Recall@1 of 52.83 and MRR of 0.650, compared to 46.75/0.567 for Qwen3-Embedding and 35.60/0.441 for E5-Mistral-7B-Instruct.
>
>         Overall, in the joint embedding space learned by ProofBridge, equivalent NL-FL pairs form tight clusters, while inequivalent pairs remain well separated, enabling reliable retrieval of the most semantically relevant FL proofs (demonstrations) for conditioning the LLM during auto-formalization.
>
> _[continued in next comment box 2/3...]_

---

> ### Author Response · Authors · 2025-11-22
> **Authors' response to Reviewer B6B5 (Part 2/3)**
>
> _[...continued from previous comment box 1/3]_
>
> - **Comparison with the NL theorem retrieval mechanism in “Herald: A Natural Language Annotated Lean 4 Dataset”[2].** HERALD uses the same LeanSearch[1] tool, where the task is FL theorem to NL theorem retrieval. As in LeanSearch, only theorem statements are retrieved, and proofs are not part of the retrieval objective. HERALD also uses the same SoTA text encoder (E5-Mistral-7B-Instruct), and our earlier arguments and empirical findings apply directly here.
>
>     * Importantly, retrieving FL proofs is a fundamentally different and more challenging problem. Raw-text encoders such as E5-Mistral-7B-Instruct rely heavily on surface-level lexical cues (keywords). In Lean, many proofs share similar tactics and token patterns, so keyword overlap is not discriminative. As shown in our experiments, this leads to poor separation between retrieved and non-retrieved instances in the embedding space and consequently weak FL theorem+proof retrieval performance.
>
>     * In contrast, ProofBridge incorporates the DAG structure of proofs and learns a joint NL to FL alignment that captures fine-grained proof semantics, which enables efficient retrieval of semantically-relevant theorem+proof pairs rather than theorems (statements) alone.
>
> - **Comparison with the FL theorem retrieval mechanism in “Rethinking and Improving Autoformalization…”[3].** This work retrieves FL theorems given an NL theorem as input. As with LeanSearch and HERALD, the retrieval target is only the theorem (statement), and the FL proofs are neither stored nor retrieved. Retrieving proofs is a different problem, and our empirical results show that encoding Lean proofs as plain text is not effective for distinguishing among them.
>
> ------
>
> **_[W3] The "Semantic Correctness" (SC) evaluation metric is not new. In [1], the BEq metric was introduced ... basically the same as SC. Subsequent work such as RLMEval [2] and StepFun-Formalizer [3] have extended and improved BEq … would encourage the authors to recognize the previous work._**
>
> We thank the reviewer for pointing out BEq and related prior work on semantic correctness metrics. We have incorporated the reviewer’s feedback in the **revised submission** by adding the relevant citations and discussing key differences between them and our Semantic Correctness (SC) metric (_Section A.4 in the Appendix_).
>
> While the general idea behind our SC metric and BEq[1,3] is similar, as both aim to establish bidirectional equivalence in Lean, the sets of allowed tactics differ. Since the notion of equivalence depends on the permitted tactics, these differences lead to meaningful distinctions between SC and BEq. BEq+[2] is a reference-based metric inspired by BEq. However, BEq+ is deterministic and CPU-efficient, while SC relies on an LLM-based proof synthesizer. This creates a trade-off: the LLM judge can capture equivalences beyond the reach of the deterministic procedure, whereas BEq+ provides a reproducible evaluation.
>
> To illustrate the difference between our SC metric and BEq, we refer the reviewer to the example provided in **Appendix A.4** (_page 16_). In the appendix example, the two propositions differ primarily in the use of an auxiliary variable (v):
>
> - Proposition 1 (gold-standard FL theorem from miniF2F-Test-PF) explicitly introduces an auxiliary variable v to denote volume.
>
> - Proposition 2 (produced by Kimina-Prover-RL-1.7B) omits the auxiliary variable and substitutes the corresponding formula directly.
>
> To the best of our understanding of BEq as presented in the paper[1], these propositions would not be recognized as equivalent under the tactic set permitted by BEq. Our SC metric, by contrast, was specifically designed to handle such forms of equivalence. Variations like this occur frequently in practice, and human experts themselves often differ in whether they prefer introducing auxiliary variables. By explicitly handling these variations and using LLM-generated bidirectional proofs that are type-checked, SC provides an evaluation that is both more lenient in accepting semantically equivalent variations and more faithful in assessing the performance of autoformalization models.
>
> Further, we have added a **qualitative example** in _Appendix A.5_ of the revised submission. The example presents an NL theorem and proof pair from miniF2F-Test-PF, showing: (i) the pass@1 proof auto-formalization output from Kimina-Prover-RL-1.7B in the text-based retrieval few-shot setting, which is type-correct but not semantically correct, (ii) the retrieved semantically relevant formal Lean theorem and proof pairs from our database, and (iii) the pass@1 proof auto-formalization output from ProofBridge, which is both type-correct and semantically correct (along with the proof of bi-directional equivalence to the gold-standard theorem).
>
> ------
> _[continued in next comment box 3/3...]_

---

> > ### Author Response · Authors · 2025-11-22
> > **Authors' response to Reviewer B6B5 (Part 3/3)**
> >
> > _[...continued from previous comment box 2/3]_
> >
> > **_[W4] …retriever baseline, all-MiniLM-L6-v2, shouldn't be considered a SOTA…a small model pretrained on general NL data … try a larger pretrained model such as e5-mistral-7b-instruct (which LeanSearch seems to use), or at least not label the current baseline as SOTA (because LeanSearch exists)._**
> >
> > We have incorporated the reviewer’s feedback and added new empirical comparisons in the **revised submission** for NL/FL Cross-Modal Retrieval Performance (Table 1). In the revised submission, we compare ProofBridge’s joint-embedding model against encoding NL and FL theorem-proof pairs using three raw-text encoders (_detailed in Section 5.3_):
> >
> > **(a)** all-MiniLM-L6-v2 (Baseline Encoder, 22.7M parameters)
> >
> > **(b)** Qwen3-Embedding-8B (SoTA Encoder, as suggested by Reviewer 9ttn)
> >
> > **(c)** E5-Mistral-7B-Instruct (SoTA Encoder, as suggested by Reviewer B6B5)
> >
> > We also introduce a **new metric**, mean Median Gap (mMG), which quantifies the separation between retrieved and non-retrieved instances (_detailed in Section 5.2_). We present the key findings below:
> >
> > 1. Despite being ~32× smaller than SoTA encoders, ProofBridge outperforms all SoTA and Baseline Encoders in Recall, MRR, and mMG (_Section 5.5_).
> >
> > 2. Plain-text encoders collapse many mathematical texts into similar embedding regions and rely on lexical similarity (keywords), which is ineffective for Lean proofs which share multiple same tactics.
> >
> > 3. ProofBridge leverages DAG-aware proof encoding and contrastive joint NL-FL embeddings, capturing fine-grained proof semantics, which:
> >
> >     - Increase separation between retrieved and non-retrieved instances in the embedding space
> >
> >     - Enable effective retrieval of semantically relevant theorem+proof pairs to condition LLMs during proof auto-formalization
> >
> > We refer the reviewer to our earlier response to W-1/W-2 and to _Section 5.5_ and _Table 1_ in the **revised submission** for a detailed empirical comparison with the SoTA Encoders.
> >
> > ------
> >
> > **_[Q1] For the same NL problem, there might exist different correct ways to state it in Lean, some being easier to prove than others. Do the authors envision ProofBridge to be extensible to optimize for downstream provability of formalized statements?_**
> >
> > We thank the reviewer for the insightful question. ProofBridge is primarily designed for faithful NL-to-FL autoformalization, aligning theorems+proof pairs in a joint NL-FL semantic space and leveraging retrieval-augmented fine-tuning with iterative proof repair. This architecture naturally supports optimizing downstream provability by incorporating feedback from Lean’s type checker and bidirectional semantic equivalence. Due to the flexibility of our loss formulation, ProofBridge can be extended by adjusting the loss or weighting retrieved examples to prefer FL representations that are easier to prove. This extension would be straightforward to implement within our existing retrieval-augmented fine-tuning framework.
> >
> > ------

---

### Official Review · Reviewer_h6DL · 2025-10-27

**Soundness:** 2
**Presentation:** 3
**Contribution:** 2
**Rating:** 2
**Confidence:** 4

**Summary:**

This paper presents a framework for automatically translating natural language mathematical theorems and proofs into Lean 4 formal language with RAG. The approach consists of three main components: (1) a joint embedding model that aligns natural and formal language theorem-proof pairs, (2) retrieval-augmented fine-tuning of LLMs, and (3) an iterative repair mechanism that uses Lean's type checker and semantic equivalence verification. The authors demonstrate improvements over baselines on some experiments.

**Strengths:**

1. The use of LeanDojo’s ByT5 encoder to capture the "linearized DAG traversal" of Lean proofs, rather than just treating the code as flat text. It might be a thoughtful method for encoding the structural properties of formal proofs.
2. The integration of RAG and SFT is well-executed. The authors provide corresponding ablation studies which help demonstrate the impact of SFT.

**Weaknesses:**

1. The core motivation of the task, i.e., translating a known natural language proof into a formal proof, is questionable. The primary value of formal math languages is to verify statements whose correctness is uncertain. In most practical scenarios, a formal statement is the starting point, and the proof is the unknown to be found. If an informal proof is already known to be correct, the motivation for formalization becomes unclear. Of course, you can state that it's useful in some scenarios, like translating textbooks and formalizing existing proofs. But currently, what you are claiming is that the previous 2-stage practice (formalize statement and prove in formal language) is problematic, and yours is correct. It's not the case.
2. The paper claims the superiority of encoding the "linearized DAG traversal" using the ByT5 encoder. However, there is no ablation study to support this. How does this compare to a simpler baseline, such as encoding the raw text of the FL proof?
3. The experimental comparisons in Table 2 are insufficient and potentially misleading: (1) The baseline models were also provided with the full input context, i.e., the NL theorem, the NL proof, and retrieved RAG examples, right? Then the authors should provide ablations that isolate the benefit of the (arguably unrealistic) input. Specifically, how much does adding the NL proof and the RAG component each contribute to the performance of both the baseline and the proposed model? This is crucial to understand if PROOFBRIDGE is genuinely better at using the informal proof, or whatever else. (2) The reported Semantic Correctness for powerful models like Gemini-2.5-Pro and GPT-5-mini seems anomalously low. The authors should provide a case study or a deeper analysis explaining these failures. (3) The paper dismisses Kimina-Autoformalizer for achieving 0% correctness. This is a strawman argument. These models are designed for statement formalization only and to output sorry, not end-to-end proof generation. Stating they "perform poorly" is an unfair characterization of models built for a different task.
4. The work is missing a comparison to REAL-Prover: Retrieval Augmented Lean Prover for Mathematical Reasoning (arXiv:2505.20613, 2025), which is highly relevant to this work.

**Questions:**

Please answer the question in Weakness.

---

> ### Author Response · Authors · 2025-11-22
> **Authors' response to Reviewer h6DL (Part 1/4)**
>
> We sincerely thank the reviewer for their thoughtful feedback and constructive suggestions. We also appreciate the valuable time spent preparing a detailed review. In response, we have revised the paper to include new comparative results against SoTA models, more detailed ablation studies, and a qualitative example. Below, we address the comments, organized under Weaknesses (W) and Questions (Q). We hope that these changes make the paper more thorough and comprehensive, and address the reviewer’s concerns.
>
> ------
>
> **_[W1] The core motivation of the task, i.e., translating a known natural language proof into a formal proof, is questionable…In most practical scenarios, a formal statement is the starting point, and the proof is the unknown to be found…If an informal proof is already known to be correct, the motivation for formalization becomes unclear…_**
>
> We sincerely thank the reviewer for raising this concern. We respectfully disagree with the assertion that translating natural language (NL) proofs to formal language (FL) proofs is of limited value. Our clarification of the motivation for proof auto-formalization is as follows:
>
> - **NL proofs are never guaranteed to be perfectly correct due to their inherent ambiguity.** By definition, any proof written in NL is ambiguous; it can contain subtle errors or hidden assumptions, even when it appears detailed. As discussed in the Introduction (Section 1), many historical counterexamples illustrate this, including cases where seemingly correct NL proofs were later found to contain non-trivial errors. Only by formalizing NL proofs into languages such as Lean, Isabelle, or Coq, each with strict syntax and symbolic logic, can we achieve reliable automated verification guarantees. Thus, the process of proof auto-formalization resolves the inherent ambiguity of NL proofs via computer-verifiable FL proofs.
>
> - **Proof discovery is a separate problem; in practical scenarios, efficient FL proof synthesis relies on guidance from NL proofs or proof sketches.** We respectfully note that our work focuses on proof auto-formalization, not on proof discovery. Proof discovery is a distinct problem: given an FL theorem, the goal is to generate a correct FL proof without any guidance, as for some open problems even mathematicians may not know how the proof should proceed. In such cases, the tool must search blindly (inefficiently or using heuristics) over a vast space of tactics.
>
>     However, for the majority of mathematical problems, humans typically have an idea of the proof in NL, or at least a proof sketch, though its correctness is not guaranteed until it is formalized. Proof auto-formalization leverages this guidance, accelerating FL proof construction and helping uncover hidden errors in the NL proof or sketch. In practice, almost all human formalization efforts, such as in Lean, Isabelle, or Coq, begin by assuming the NL proof is correct and validate it through translation to FL. Attempting to generate FL proofs “from nothing” is rarely practical; using NL proofs as a starting point is both realistic and essential for efficient formalization. This approach underlies interactive theorem proving (ITP), where humans provide strategies and the system verifies them.
>
>     Lastly, starting from an FL theorem necessitates human hand-formalization of the NL theorem, which impedes a fully automated formalization pipeline.
>
> - **NL proofs provide valuable structural guidance, even if imperfect or incomplete.** Even if an NL proof contains minor errors, it can still be highly useful for a proof auto-formalization tool in planning the structure of the corresponding FL proof, for example, indicating whether induction, case analysis, or proof by contradiction should be applied. In the rare case where an NL proof is completely incorrect, auto-formalization may fail. Similar failures can also occur in proof discovery when no guidance is provided. These cases highlight an opportunity for future research: analyzing whether the failure arises from limitations of the prover or flaws in the NL proof itself. The ability to validate and refine the correctness of arbitrary NL proofs is made possible only through proof auto-formalization.
>
> Overall, we believe that a system translating NL theorem-proof pairs into FL is far more practical and usable than hand-formalizing the theorem first, and then relying on proof discovery, particularly when humans can provide guidance through an NL proof sketch. Importantly, we are not claiming that the traditional two-stage approach of “formalize the statement, then search for a formal proof from scratch” is invalid or problematic. Rather, we argue that leveraging NL proof guidance provides a complementary and often highly effective signal, especially for complex problems where formal proof search alone may be insufficient. Our framework is designed to augment existing theorem-proving approaches, not replace them.
>
> ------
> _[continued in next comment box 2/4...]_

---

> > ### Author Response · Authors · 2025-11-22
> > **Authors' response to Reviewer h6DL (Part 2/4)**
> >
> > _[...continued from previous comment box 1/4]_
> >
> > **_[W2] …paper claims the superiority of encoding the "linearized DAG traversal" using the ByT5 encoder … no ablation study to support this … how does it compare to encoding raw text of the FL proof?_**
> >
> > **Firstly**, in the **revised submission**, we have incorporated the reviewer’s feedback and added the ablation studies in _Section 5.6_. Specifically, alongside ProofBridge (built upon Kimina-Prover-RL-1.7B and using our DAG-aware encoder for retrieving examples during both fine-tuning and inference), we now compare three Kimina-Prover variants (RL-1.7B, Distill-8B, and 72B) under the three experimental settings described in _Section 5.4_:
> >
> > **(a)** **Zero-shot:** with no in-context I/O examples;
> >
> > **(b)** **Random few-shot:** with five randomly selected I/O examples; and
> >
> > **(c)** **Text-based retrieval few-shot:** where the top-5 FL theorem-proof pairs are retrieved from $\mathcal{D}$ using Qwen3-Embedding-8B and included with their NL counterparts as in-context I/O examples.
> >
> > From the pass@32 results in _Table 2_, Kimina-Prover-RL-1.7B achieves 40.16% SC and 75.00% TC in the zero-shot setting. When random examples are added, SC drops to 31.56% while TC rises to 93.85%, with similar trends across variants. This indicates that random examples improve TC (syntax) but hurt SC by causing the model to hallucinate semantically misaligned proofs. With text-based retrieval via Qwen3-Embedding-8B, SC rises to 32.38% but TC declines, potentially because QA-based retrieval favors proofs with similar tactics, reducing tactic diversity. This highlights the need for retrieving semantically relevant examples via a DAG-aware encoder, as in ProofBridge. (_detailed in Section 5.6_)
> >
> > **Secondly**, in _Section 5.5_ and _Table 1_ of the **revised submission**, we compare ProofBridge’s joint-embedding model against encoding NL and FL theorem-proof pairs using three raw-text encoders:
> >
> > - all-MiniLM-L6-v2 (Baseline Encoder, 22.7M parameters)
> >
> > - Qwen3-Embedding-8B (SoTA Encoder, as suggested by Reviewer 9ttn)
> >
> > - E5-Mistral-7B-Instruct (SoTA Encoder, as suggested by Reviewer B6B5)
> >
> > We also introduce a **new metric**, mean Median Gap (mMG), which quantifies how well the retrieved and non-retrieved instances are separated in the embedding space. (_detailed in Section 5.2_)
> >
> > As shown in _Section 5.5_, ProofBridge outperforms both the Baseline and SoTA encoders in Recall, MRR, and mMG, despite being roughly 32x smaller than the SoTA. We attribute this to two key factors:
> >
> > - **QA-oriented, text-based encoders capture only coarse domain signals.** These models are trained on broad corpora (e.g., legal, medical), causing many mathematical texts to collapse into similar regions of the embedding space. In Table 2, for E5-Mistral-7B-Instruct and Qwen3-Embedding-8B in NL-to-FL retrieval, the median cosine similarity of top-K retrieved examples is very close to that of non-retrieved ones (mMG = 0.10 and 0.29, respectively). In contrast, ProofBridge achieves a significantly larger separation (mMG = 0.65).
> >
> > - **SoTA text encoders depend heavily on lexical similarity, ineffective for Lean proofs.** Raw-text encoders often rely on superficial lexical cues such as keywords. In Lean, many proofs use the same tactics and similar surface tokens, so keyword overlap becomes non-discriminative. ProofBridge instead incorporates the DAG structure of proofs and learns a contrastive joint alignment using a DAG-aware FL encoder that captures fine-grained proof semantics. For NL-to-FL retrieval, ProofBridge achieves Recall@1 of 52.83 and MRR of 0.650, compared to 46.75/0.567 for Qwen3-Embedding and 35.60/0.441 for E5-Mistral-7B-Instruct.
> >
> > Overall, in the joint embedding space learned by ProofBridge, equivalent NL-FL pairs form tight clusters, while inequivalent pairs remain well separated, enabling reliable retrieval of the most semantically relevant FL proofs (demonstrations) for conditioning the LLM during auto-formalization.
> >
> > ------
> > _[continued in next comment box 3/4...]_

---

> > > ### Author Response · Authors · 2025-11-22
> > > **Authors' response to Reviewer h6DL (Part 3/4)**
> > >
> > > _[...continued from previous comment box 2/4]_
> > >
> > > **_[W3-partA] …provide ablations that isolate the benefit of the input…how much does adding the NL proof and the RAG component each contribute ... crucial to understand if ProofBridge is genuinely better at using the informal proof…_**
> > >
> > > In the **revised submission**, as detailed in our response to [W2], to assess the contribution of in-context examples, we compare three SoTA Kimina-Prover variants (RL-1.7B, Distill-8B, and 72B) under the three experimental settings described in **Section 5.4**: zero-shot, random few-shot, and text-based retrieval few-shot. The zero-shot setting uses no in-context I/O examples; the random few-shot setting includes five randomly selected I/O examples; and the text-based retrieval few-shot setting includes the top-5 FL theorem-proof pairs retrieved using Qwen3-Embedding-8B (SoTA Encoder), together with their NL counterparts, as in-context I/O examples.
> > >
> > > ProofBridge is built on top of Kimina-Prover-RL-1.7B. To isolate the contribution of the RAG component, **Section 5.6** reports an ablation across three variants:
> > >
> > > - **ProofBridge (SFT)**, which is fine-tuned only on labeled NL–FL pairs and does not incorporate retrieved proofs during fine-tuning, improves SC by +2.06%.
> > >
> > > - **ProofBridge (Retrieval-augmented SFT)**, which fine-tunes the LLM with semantically relevant retrieved FL proofs included in the input, improves SC by +22.55%.
> > >
> > > - **ProofBridge (Retrieval-augmented SFT + Repair)**, which further adds iterative proof-repair during inference, improves SC by +29.92%.
> > >
> > > These ablations directly quantify the value added by the NL proof (via in-context demonstrations) and by the RAG component, demonstrating that ProofBridge leverages informal proof guidance far more effectively than baseline models.
> > >
> > > ------
> > >
> > > **_[W3-partB] …reported Semantic Correctness for powerful models like Gemini-2.5-Pro and GPT-5-mini seems anomalously low. ... provide a case study or a deeper analysis ..._**
> > >
> > > The low SC of foundation models such as Gemini-2.5-Pro and GPT-5-mini can be attributed to several factors. Although these models excel at NL generation, they are not trained to strictly adhere to the syntactic and semantic rules of specialized formal languages (FLs) like Lean. Producing a valid formal proof requires precise handling of proof tactics, dependency graphs, and formal logic structures, which are often underrepresented or absent in the models’ pretraining corpora. We have added this line of reasoning in **Section 5.5** of the **revised submission** when presenting the results of foundation models.
> > >
> > > When formalizing NL proofs from miniF2F-Test-PF, Gemini-2.5-Pro frequently generated sequences superficially resembling Lean syntax but with mismatched tactics, missing arguments, or invalid references to definitions. GPT-5-mini often produced syntactically invalid Lean tokens and violated key dependencies in the proof DAG, resulting in verification failures. These demonstrate that even powerful LLMs cannot reliably perform AFPS or full proof auto-formalization out-of-the-box. As reflected in _Table 2_, these models’ TC accuracies are low, which directly limits SC, as semantic correctness requires type-correct proofs.
> > >
> > > Because foundation models are not inherently effective for AFPS and auto-formalization in specialized FLs like Lean, several SoTA LLMs have been developed for these tasks. We list and compare many such tools in Section 5.3 and evaluate ProofBridge against them in _Table 1_ and _Section 5.5_.
> > >
> > > ------
> > >
> > > **_[W3-partC] …paper dismisses Kimina-Autoformalizer for achieving 0% correctness…these models are designed for statement formalization only and to output sorry, not end-to-end proof generation. Stating they "perform poorly" is an unfair characterization of models built for a different task._**
> > >
> > > We acknowledge that Kimina-Autoformalizer-7B and Herald-Translator are designed specifically for theorem (statement) auto-formalization. Our rationale for including these models is that, since our work addresses end-to-end auto-formalization (NL-to-FL), it is important to compare against available SoTA auto-formalization tools. Our results show that models focused solely on statement formalization are unsuitable for generating full FL proofs. In contrast, automated formal proof synthesis (AFPS) tools are better suited for proof auto-formalization, as they are trained to generate complete FL proofs with tactics, which is the same output space targeted in proof auto-formalization.
> > >
> > > Having said that, we agree that “perform poorly” might be an unfair characterization. We have therefore incorporated the reviewer’s suggestions and updated **Section 5.5** in the **revised submission** to clarify: _“These models are designed to formalize theorem statements only, leaving proofs as sorry. They lack knowledge of proof DAGs and tactics, making them unsuitable for end-to-end proof auto-formalization.”_
> > >
> > > ------
> > > _[continued in next comment box 4/4...]_

---

> > > > ### Author Response · Authors · 2025-11-22
> > > > **Authors' response to Reviewer h6DL (Part 4/4)**
> > > >
> > > > _[...continued from previous comment box 3/4]_
> > > >
> > > > **_[W4] The work is missing a comparison to REAL-Prover: Retrieval Augmented Lean Prover for Mathematical Reasoning (arXiv:2505.20613, 2025), which is highly relevant to this work._**
> > > >
> > > > We thank the reviewer for pointing out REAL-Prover. To address the reviewer’s concern in the **revised submission**, we have cited REAL-Prover in **Section 2 (Related Work)** and highlighted the distinctions with our approach. REAL-Prover implements a next-tactic prediction (NTP) strategy, where the LLM is invoked stepwise conditioned on the current proof state, augmented by retrieving relevant Mathlib theorems via LeanSearch-PS. It projects proof states and theorems into a shared semantic space for nearest-neighbor retrieval.
> > > >
> > > > We distinguish ProofBridge from REAL-Prover along two key lines:
> > > >
> > > > - First, while REAL-Prover focuses on stepwise tactic prediction, ProofBridge adopts an auto-formalization as well as a whole-proof generation (WPG) approach, generating complete FL proofs in a single pass. Recent WPG-based systems, including DeepSeek-Prover, TheoremLlama, Kimina-Prover, Goedel-Prover, Leanabell-Prover, and STP, demonstrate computational efficiency by avoiding repeated interactions inherent in stepwise NTP.
> > > >
> > > > - Second, LeanSearch-PS uses two dense retrievers trained via plain-text contrastive encoding (E5-Mistral-7B-Instruct), whereas ProofBridge employs a DAG-aware FL encoder that linearizes proof DAGs to capture structural semantics. The intuition is that semantically similar proofs, with analogous DAG structures of proof states and tactics, should produce similar embeddings. We further train a joint NL-FL embedding model with symmetric contrastive loss so that semantically equivalent pairs are close in the shared space, enabling efficient NL-to-FL cross-modal retrieval.
> > > >
> > > > Additionally, in the **revised submission** we compare against the **SoTA Encoder** used by REAL-Prover (**E5-Mistral-7B-Instruct**). In _Table 2_, top-K retrieval using E5-Mistral-7B-Instruct and Qwen3-Embedding-8B shows median cosine similarities close to those of non-retrieved examples (mMG = 0.10 and 0.29), whereas ProofBridge achieves a substantially larger separation (mMG = 0.65). This design allows equivalent NL-FL pairs to form tight clusters while inequivalent pairs remain well separated, enabling reliable retrieval of semantically relevant FL proofs for conditioning the LLM during auto-formalization. (_detailed in Section 5.5_)
> > > >
> > > > ------
> > > >
> > > > Below, we include two additional updates in the **revised submission**:
> > > >
> > > > - We have added a **qualitative example** in _Appendix A.5_ of the revised submission. The example presents an NL theorem and proof pair from miniF2F-Test-PF, showing: (i) the pass@1 proof auto-formalization output from Kimina-Prover-RL-1.7B in the text-based retrieval few-shot setting, which is type-correct but not semantically correct, (ii) the retrieved semantically relevant formal Lean theorem and proof pairs from our database, and (iii) the pass@1 proof auto-formalization output from ProofBridge, which is both type-correct and semantically correct (along with the proof of bi-directional equivalence to the gold-standard theorem).
> > > >
> > > > - We have added _Figure 3_, showing the pass@32 performance of ProofBridge and various SoTA tools **across mathematical domains**: induction/sequence, number theory, algebra, and contest problems from AIME, AMC, and IMO. ProofBridge performs best on number theory, achieving over 85% SC, while contest problems remain the most challenging, reaching about 35% SC (_detailed in Section 5.5_).
> > > >
> > > > ------

---

> > ### Comment · Reviewer_h6DL · 2025-11-22
> >
> > I recognize the value of translating NL proofs to FL proofs. What I strongly disagree with is the authors' assertion that the prevailing two-step paradigm (first translating theorems, then synthesizing proofs) creates a "fundamental disconnect" and inherently "cannot utilize the rich logic and step information contained in natural language proofs". Separating the formalization of the statement from the generation of the formal proof does not preclude the proof synthesis model from utilizing the natural language proof. In a robust two-step pipeline, the proof synthesis model can condition on both the formalized statement and the original natural language proof to guide the tactic generation. Therefore, the claim that an "end-to-end" architecture (Input: NL pair → Output: FL pair) is the only or "ideal" way to leverage the logic within NL proofs is a false dichotomy. The "disconnect" described by the authors appears to be an artificial constraint rather than an intrinsic limitation of the two-step approach.

---

> > > ### Author Response · Authors · 2025-11-26
> > > **Authors' response to Reviewer h6DL**
> > >
> > > We thank the reviewer for sharing their concern. Based on the reviewer’s feedback, we are happy to have revised the abstract and relevant sections to ensure that our wording does not imply any inherent superiority of a unified pipeline over the well-established two-step paradigm. Our intention is not to claim that end-to-end proof auto-formalization is the only or ideal way to leverage the logic present in NL proofs. We emphasize that our **unified approach is complementary to, rather than a replacement for, the two-step paradigm**. Our motivation for exploring a unified system is detailed below:
> > >
> > > - **Clarification of intent:** We completely agree with the reviewer that a well-designed two-step pipeline can condition on the NL proof when synthesizing formal proofs. Our interest in unified architectures arises from certain _practical considerations_. From our experience as users of various SoTA tools, having theorem translation and proof synthesis accessible within a **single interface can simplify interactions and reduce coordination overhead**. For instance, a two-step pipeline requires $k$ theorem translation runs and $k$ proof synthesis runs for pass@$k$, effectively doubling computation and adding coordination overhead when incorporating repair feedback from a judge (e.g., Lean type-checker, LLM judge, or a human). A unified system handles generation and repair coherently, which can be more convenient in practice.
> > >
> > > - **Empirical comparison with a strong two-step baseline:** We evaluated a **SoTA Two-Step** pipeline, using Herald-Translator for theorem translation and Kimina-Prover-Distill-8B for proof generation. Empirically, its pass@32 performance (43.44% SC, 59.43% TC) is lower than that of ProofBridge (62.70% SC, 95.49% TC). We also observed that it is susceptible to cascading errors: a semantically or type-incorrect FL theorem from the first model leads the second to produce a semantically incorrect theorem-proof pair. We describe the setup in Section 5.4 and present the results in Section 5.5 and Table 2 of the **revised PDF**.
> > >
> > > - **Core ProofBridge components applicable to two-step and unified pipelines:** Our proposed NL/FL cross-modal retrieval and retrieval-augmented SFT are **effective regardless of whether one uses a unified system or a two-step pipeline**. By supplying semantically relevant proof structures and reusable tactics, we envision that these components will improve the performance of both theorem-only auto-formalization models and AFPS LLMs.
> > >
> > > - **Unified pipelines as an underexplored, complementary direction:** Unified proof auto-formalization pipelines remain **relatively underexplored**, with only a few systems, such as Draft-Sketch-Prove (Isabelle) and FormL4 (Lean), attempting full theorem+proof auto-formalization. ProofBridge aims to broaden this design space while remaining fully interoperable with existing theorem-only auto-formalizers and AFPS LLMs.
> > >
> > > - **Clarifying our reference to AlphaProof:** Our reference to AlphaProof was meant solely to illustrate that, **in current practice, pipelines still rely on human intervention**, even though SoTA auto-formalizers and AFPS tools exist. This underscores the broader challenge of ensuring the semantic correctness of proof auto-formalization. We have refined the text to avoid suggesting that this reflects any fundamental limitation of the two-step paradigm.
> > >
> > > To address the reviewer’s concern and clarify any misunderstanding, we have revised the PDF and abstract. We sincerely hope these revisions fully address the reviewer’s concerns and clarify our intentions, and **we would be grateful for any further feedback that might help in considering an increase to the score**. We thank the reviewer again for their valuable feedback.
> > >
> > > ------
> > >
> > > The changes are summarized below.:
> > >
> > > **Abstract:** [modified text]
> > >
> > > _…Most state-of-the-art methods either focus on theorem-only NL-to-FL auto-formalization or on FL proof synthesis from FL theorems. In practice, auto-formalization of both theorem and proof still requires human intervention, as seen in AlphaProof’s silver-medal performance at the 2024 IMO, where problem statements were manually translated before automated proof synthesis. …_
> > >
> > > **Introduction (Section 1):**  [modified text]
> > >
> > > _...AlphaProof…followed this two-step process…. Thus, in practice, pipelines still require manual formalization of the theorem before proof synthesis, even though SoTA theorem-only auto-formalization and AFPS tools exist. This illustrates the broader challenge that current systems often rely on human intervention to ensure semantic correctness of proof auto-formalization._
> > >
> > > **Experimental Evaluation (Section 5):**  [added result]
> > >
> > > We have added experimental results for the _SoTA Two-Step_ setup. The setup is described in Section 5.4, the results are presented in Table 2, and the analysis is discussed in Section 5.5.
> > >
> > > ------

---

### Official Review · Reviewer_9ttn · 2025-10-31

**Soundness:** 2
**Presentation:** 1
**Contribution:** 2
**Rating:** 2
**Confidence:** 4

**Summary:**

This paper presents ProofBridge, a unified framework for the autoformalization of NL math proofs into Lean4. It contains three main components: (1) a joint embedding model that aligns NL and FL theorems in a shared semantic space; (2) a retrieval-augmented SFT that conditions proof autoformalization; (3) an iterative proof repair mechanism that combines Lean type-checking and semantic equivalence verification. The authors curate two benchmarks: NuminaMath-Lean-PF and MiniF2F-Test-PF for evaluation. Experiment results show improvements over all-MiniLM-L6-v2 on the retrieval task and Kimina-Prover-RL-1.7B in the proof autoformalization task.

**Strengths:**

1. This paper presents an interesting proof repair technique by a proof autoformalizer
2. The experiment indicates the framework performs well in the 1.7B level model
3. The NuminaMath-Lean-PF contributes a 40k-level dataset of natural language to a formal language autoformalization dataset.

**Weaknesses:**

Despite the paper’s mentioned strengths, it requires significant revision to address the following cirtical weaknesses:

1. **Overclaiming SOTA:** My primary concern is that the paper claims to outperform previous SOTA models and position itself as the new SOTA. However, the baselines it uses for comparison are clearly not current SOTA methods in either proof retrieval or proof autoformalization tasks. For the retrieval task, all-MiniLM-L6-V2 is an outdated model in 2020. More recent and competitive baselines should be considered, such as Qwen3-Embedding-8B [1]. For the proof autoformalization task, comparing solely against the 1.7B variant of Kimina-Prover is insufficient, as the same work introduces 8B and 72B versions that significantly outperform the 1.7B variant. The overclaiming of SOTA significantly harms the objectivity of the paper.
2. **Questionable Evaluation Methodology:** The evaluation procedure for proof retrieval is overly idealized and does not adequately reflect real-world complexity. The task assumes that for each NL input, there exists an exact corresponding FL proof in the dataset, and the goal is to simply retrieve this match. However, realistic scenarios are much more complex. For example, there may be many out-of-distribution queries and hard negative samples that require fine-grained discrimination.
3. **Incomplete experimental analysis:** The experimental section lacks the depth and rigor expected for the conference paper. It misses some important ablation studies, such as the impact of individual components, the effect of retrieval hyperparameters, the analysis of inference hyperparameters, and the impact of training data size and composition. It also lacks qualitative analysis, such as examples of successful and failed retrievals, a qualitative comparison between generated and gold-standard proofs, and case studies on the repair process.
4. **Poor paper organization and clarity:** The paper suffers from significant structural and clarity issues:
    1. Structural problems: The NuminaMath-Lean-PF dataset is first mentioned in the methodology section in line 304 without any introduction to its construction and characteristics. Readers must go to Section 5 to find its details.
    2. Unclear experiment section: The experiment section is difficult to read due to lack of clear subsection organization (such as separate subsections for retrieval results, autoformalization results, ablations, and additional studies). It also mixes the main results and ablation studies, and provides an insufficient description of experimental settings.

[1] Zhang, Y., Li, M., Long, D., Zhang, X., Lin, H., Yang, B., ... & Zhou, J. (2025). Qwen3 Embedding: Advancing Text Embedding and Reranking Through Foundation Models. *arXiv preprint arXiv:2506.05176*.

**Questions:**

1. Why was Kimina-Prover-RL-1.7B chosen as the primary baseline instead of larger variants?
2. What is the out-of-distribution test results or adversarial perturbations results on the retrieval task
3. What is the performance breakdown across different mathematical domains
4. Can the authors provide comprehensive qualitative examples showing when and why the framework helps?

---

> ### Author Response · Authors · 2025-11-22
> **Authors' response to Reviewer 9ttn (Part 1/4)**
>
> We sincerely thank the reviewer for their thoughtful feedback and constructive suggestions. We also appreciate the valuable time spent preparing a detailed review. In response, we have revised the paper to include new comparative results against SoTA models, more detailed ablation studies, and a qualitative example. Below, we address the comments, organized under Weaknesses (W) and Questions (Q). We hope that these changes make the paper more thorough and comprehensive, and address the reviewer’s concerns.
>
> ------
>
> **_[W1-partA] Overclaiming SOTA: …baselines it uses for comparison are clearly not current SOTA methods in either proof retrieval or proof autoformalization tasks…For the retrieval task, all-MiniLM-L6-V2 is an outdated model in 2020. More recent and competitive baselines should be considered, such as Qwen3-Embedding-8B…_**
>
> We have incorporated the reviewer’s feedback and added new empirical comparisons in the **revised submission** for NL/FL Cross-Modal Retrieval Performance (_Table 1_). In the revised submission, we compare ProofBridge’s joint-embedding model against encoding NL and FL theorem-proof pairs using three raw-text encoders (detailed in _Section 5.3_):
>
> **(a)** all-MiniLM-L6-v2 (Baseline Encoder, 22.7M parameters)
>
> **(b)** Qwen3-Embedding-8B (SoTA Encoder, as suggested by Reviewer 9ttn)
>
> **(c)** E5-Mistral-7B-Instruct (SoTA Encoder, as suggested by Reviewer B6B5)
>
> We also introduce a **new metric**, mean Median Gap (mMG), which quantifies the separation between retrieved and non-retrieved instances. (detailed in _Section 5.2_)
>
> As shown in _Section 5.5_, ProofBridge outperforms both the Baseline and SoTA encoders in Recall, MRR, and mMG, despite being roughly 32x smaller than the SoTA. We attribute this to two key factors:
>
> **(a) QA-oriented, text-based encoders capture only coarse domain signals.** These models are trained on broad corpora (e.g., legal, medical), causing many mathematical texts to collapse into similar regions of the embedding space. In Table 2, for E5-Mistral-7B-Instruct and Qwen3-Embedding-8B in NL-to-FL retrieval, the median cosine similarity of top-K retrieved examples is very close to that of non-retrieved ones (mMG = 0.10 and 0.29, respectively). In contrast, ProofBridge achieves a significantly larger separation (mMG = 0.65).
>
> **(b) SoTA text encoders depend heavily on lexical similarity, ineffective for Lean proofs.** Raw-text encoders often rely on superficial lexical cues such as keywords. In Lean, many proofs use the same tactics and similar surface tokens, so keyword overlap becomes non-discriminative. ProofBridge instead incorporates the DAG structure of proofs and learns a contrastive joint alignment using a DAG-aware FL encoder that captures fine-grained proof semantics. For NL-to-FL retrieval, ProofBridge achieves Recall@1 of 52.83 and MRR of 0.650, compared to 46.75/0.567 for Qwen3-Embedding and 35.60/0.441 for E5-Mistral-7B-Instruct.
>
> Overall, in the joint embedding space learned by ProofBridge, equivalent NL-FL pairs form tight clusters, while inequivalent pairs remain well separated, enabling reliable retrieval of the most semantically relevant FL proofs (demonstrations) for conditioning the LLM during auto-formalization.
>
> ------
>
> _[continued in next comment box 2/4...]_

---

> > ### Author Response · Authors · 2025-11-22
> > **Authors' response to Reviewer 9ttn (Part 2/4)**
> >
> > _[...continued from previous comment box 1/4]_
> >
> > **_[W1-partB] Overclaiming SOTA: …For the proof autoformalization task, comparing solely against the 1.7B variant of Kimina-Prover is insufficient, as the same work introduces 8B and 72B versions that significantly outperform the 1.7B variant…_**
> >
> > We have incorporated the reviewer’s feedback and added new empirical comparisons in the **revised submission** for Proof Auto-Formalization Performance (_Table 2_). In the revised submission, we now compare **three Kimina-Prover variants** (Kimina-Prover-RL-1.7B, Kimina-Prover-Distill-8B, and Kimina-Prover-72B) under the three experimental settings described in _Section 5.4_:
> >
> > **(a)** **Zero-shot:** with no in-context I/O examples;
> >
> > **(b)** **Random few-shot:** with five randomly selected I/O examples; and
> >
> > **(c)** **Text-based retrieval few-shot:** where the top-5 FL theorem-proof pairs are retrieved from $\mathcal{D}$ using Qwen3-Embedding-8B and included with their NL counterparts as in-context I/O examples.
> >
> > These models are trained for AFPS (automated formal proof synthesis), and we observe that their relative performance differences on proof auto-formalization are small. All variants follow similar patterns of performance change across settings. From the pass@32 results in Table 2, Kimina-Prover-RL-1.7B achieves 40.16% SC and 75.00% TC in the zero-shot setting. When random examples are added, SC drops to 31.56% while TC rises to 93.85%, with similar trends across variants. This indicates that random examples improve TC (syntax) but hurt SC by causing the model to hallucinate semantically misaligned proofs. With text-based retrieval via Qwen3-Embedding-8B, SC rises to 32.38% but TC declines, potentially because QA-based retrieval favors proofs with similar tactics, reducing tactic diversity. This highlights the need for retrieving semantically relevant examples via a DAG-aware encoder, as in ProofBridge. (_detailed in Section 5.6_)
> >
> > Further, the newly added results show that ProofBridge outperforms even the 72B model in all three settings. For example, Kimina-Prover-72B achieves 44.26% SC (pass@32) under text-based retrieval few-shot, whereas ProofBridge (Retrieval-augmented SFT + Repair) reaches 62.70% SC.
> >
> > ------
> >
> > **_[W2] Questionable Evaluation Methodology: The evaluation procedure for proof retrieval is overly idealized and does not adequately reflect real-world complexity … assumes that for each NL input, there exists an exact corresponding FL proof in the dataset, and the goal is to simply retrieve this match. However, realistic scenarios are much more complex._**
> >
> > We respectfully disagree with the reviewer’s interpretation and believe there may be a misunderstanding regarding the core motivation of our approach. Our method does **not** assume that an exact FL proof exists for a given NL proof in the database. If that were the case, simple retrieval would suffice, and auto-formalization would be unnecessary. Instead, the retrieval step identifies _semantically relevant FL proofs_ that serve as _demonstrations_ to guide the downstream auto-formalizer LLM in _generating_ the FL (Lean) theorem+proof pair for a given NL input. These examples provide _cues_ about DAG structures, key tactics, and proof strategies, reflecting similar proof patterns and mathematical domains. This allows the auto-formalizer to construct correct FL proofs efficiently when no exact match exists. We also provide relevance scores for the retrieved examples, enabling the auto-formalizer LLM to weight them appropriately.
> >
> > During evaluation on miniF2F-Test-PF, we specifically ensured that formalizations for none of the input NL theorem+proof pairs exist in the database, so all results are on out-of-distribution queries. In this way, retrieval provides meaningful structural and semantic guidance rather than serving as the answer itself, addressing realistic scenarios rather than relying on perfect one-to-one correspondences.
> >
> > ------
> > _[continued in next comment box 3/4...]_

---

> > > ### Author Response · Authors · 2025-11-22
> > > **Authors' response to Reviewer 9ttn (Part 3/4)**
> > >
> > > _[...continued from previous comment box 2/4]_
> > >
> > > **_[W3]  Incomplete experimental analysis … misses some important ablation studies, such as the impact of individual components … lacks qualitative analysis, such as examples of successful and failed retrievals, a qualitative comparison between generated and gold-standard proofs, and case studies on the repair process._**
> > >
> > > We have incorporated the reviewer’s feedback in the **revised submission**:
> > >
> > > **(a)** First, we present a **dedicated sub-section (Section 5.6) for ablation studies** in the main paper, examining the impact of key individual components of ProofBridge: SFT, NL/FL cross-modal retrieval, and repair.
> > >
> > > **(b)** Second, in **Section 5.6**, to assess the effect of in-context examples, Table 2 reports zero-shot, random few-shot, and text-based retrieval few-shot performance for three Kimina-Prover variants (72B, Distill-8B, and RL-1.7B).
> > >
> > > **(c)** Third, we added **Figure 3**, showing the pass@32 performance of ProofBridge and various SoTA tools across mathematical domains: induction/sequence, number theory, algebra, and contest problems from AIME, AMC, and IMO.
> > >
> > > **(d)** Fourth, the experimental settings are detailed in Sub-section 5.4, with a cross-reference to **Appendix A.2** and **A.3** for implementation details and training hyperparameters, respectively.
> > >
> > > **(e)** Fifth, we have added a qualitative example in **Appendix A.5**, illustrating the retrieval process. Specifically, it presents an NL theorem and proof pair from miniF2F-Test-PF, showing: (i) the pass@1 proof auto-formalization output from Kimina-Prover-RL-1.7B in the text-based retrieval few-shot setting, which is type-correct but not semantically correct, (ii) the retrieved semantically relevant formal Lean theorem and proof pairs from our database, and (iii) the pass@1 proof auto-formalization output from ProofBridge, which is both type-correct and semantically correct (along with the proof of bi-directional equivalence to the gold-standard theorem).
> > >
> > > ------
> > >
> > > **_[W4-partA]  Poor paper organization and clarity … Structural problems … NuminaMath-Lean-PF dataset is first mentioned in the methodology section… without any introduction to its construction and characteristics. Readers must go to Section 5 to find its details._**
> > >
> > > We acknowledge the reviewer’s concern about the placement of dataset details. The characteristics of the training and validation dataset are first introduced in the second point of our Contributions (_Section 1, page 3_). We then briefly reference the dataset in the methodology (_Section 4, page 6_) and provide the full construction details in _Section 5 (page 7)_, which is dedicated to experimental setup and evaluation.
> > >
> > > Our intention was to keep the methodology section focused on the core proposed approach without interrupting it with implementation details. We apologize to the reviewer if this organizational choice caused any confusion. In the **revised submission**, _Section 5.1_ now clearly describes the dataset and its preparation, and we have added explicit cross-references to _Section 5.1_ in both _Section 1_ and _Section 4_.
> > >
> > > ------
> > >
> > > **_[W4-partB]  Poor paper organization and clarity … Unclear experiment section … The experiment section is difficult to read due to lack of clear subsection organization … mixes the main results and ablation studies, and provides an insufficient description of experimental settings._**
> > >
> > > In the **revised submission**, we have reorganized _Section 5 (Experimental Evaluation)_ into well-defined sub-sections:
> > >
> > > - (Section 5.1) Datasets and Preparation: NuminaMath-Lean-PF and miniF2F-Test-PF
> > >
> > > - (Section 5.2) Evaluation Metrics
> > >
> > > - (Section 5.3) State-of-the-Art Baselines (with separate paragraph headings for NL/FL cross-modal retrieval and proof auto-formalization)
> > >
> > > - (Section 5.4) Training and Evaluation Setup
> > >
> > > - (Section 5.5) Experimental Results (with separate paragraph headings for NL/FL cross-modal retrieval and proof auto-formalization)
> > >
> > > - (Section 5.6) Ablation Studies (analyzing (i) the effect of in-context examples under zero-shot, random few-shot, and text-based retrieval few-shot settings, and (ii) the contribution of each component in ProofBridge.)
> > >
> > > The experimental settings are described in Section 5.4, with implementation details and training hyperparameters provided in Appendices A.2 and A.3, respectively. In addition, Appendix A.5 includes a qualitative example, which is also referenced in Section 5.4. We sincerely hope that this restructuring addresses the reviewer’s concerns and makes the experimental section clearer and easier to follow.
> > >
> > > ------
> > > _[continued in next comment box 4/4...]_

---

> > > > ### Author Response · Authors · 2025-11-22
> > > > **Authors' response to Reviewer 9ttn (Part 4/4)**
> > > >
> > > > _[...continued from previous comment box 3/4]_
> > > >
> > > > **_[Q1]  Why was Kimina-Prover-RL-1.7B chosen as the primary baseline instead of larger variants?_**
> > > >
> > > > In the **revised submission**, we now include results for all three Kimina-Prover variants: Kimina-Prover-RL-1.7B, Kimina-Prover-Distill-8B, and Kimina-Prover-72B. The Distill-8B model is a distilled version of the 72B model, and the RL-1.7B model is the RL-tuned version of the 1.7B distilled model. All variants are evaluated in three settings: zero-shot, random few-shot, and text-based retrieval few-shot using top-5 FL theorem-proof pairs retrieved via Qwen3-Embedding-8B (detailed in Section 5.4).
> > > >
> > > > These models are trained for AFPS (automated formal proof synthesis), and we observed that their zero-shot performance differences on proof auto-formalization are relatively less. We therefore selected the RL-1.7B model for fine-tuning ProofBridge, as fine-tuning larger models is substantially more resource-intensive. That said, our approach is **model-agnostic**: the retrieved FL demonstrations and the DAG structures or lemma-usage context they provide can enhance any existing auto-formalization LLM through retrieval-augmented SFT and repair.
> > > >
> > > >  The newly added results (_Section 5.5, Table 2_) show that ProofBridge outperforms even the 72B model in all settings. For example, Kimina-Prover-72B achieves 44.26% SC (pass@32) under text-based retrieval few-shot, whereas ProofBridge (Retrieval-augmented SFT + Repair) reaches 62.70% SC.
> > > >
> > > > ------
> > > >
> > > > **_[Q2] Breakdown of errors by type (syntax errors, wrong theorem…)?_**
> > > >
> > > > Our evaluation metrics are explicitly designed to provide exactly this breakdown:
> > > >
> > > > **1. Type Correctness (TC)** measures whether the generated theorem+proof is syntactically valid i.e., successfully accepted by Lean’s type checker.
> > > >
> > > > **2. Semantic Correctness (SC)** is computed only on type-correct generations and measures whether the generated theorem is bi-directionally equivalent to the gold-standard theorem.
> > > >
> > > > Thus, all results in Table 2 of the paper report:
> > > >
> > > > - what percentage of the generations are syntactically valid (TC), and
> > > >
> > > > - what percentage corresponds to the correct theorem (SC)
> > > >
> > > > For instance, under pass@32, ProofBridge achieves 95.49% TC (syntactically valid) and 62.70% SC (syntactically valid and semantically correct), while Kimina-Prover-72B (text-based retrieval few-shot) attains 88.93% TC and 44.26% SC.
> > > >
> > > > In the **revised submission**, we additionally include _Figure 3_, which visualizes this error decomposition across mathematical domains via stacked bar plots, showing TC-SC (type-correct but semantically wrong theorem) and SC (type-correct and correct theorem) for ProofBridge and multiple SoTA baselines.
> > > >
> > > > ------
> > > >
> > > > **_[Q3]  Performance in various math domains_**
> > > >
> > > > In the **revised submission**, we added _Figure 3_, showing the pass@32 performance of ProofBridge and various SoTA tools across mathematical domains: induction/sequence, number theory, algebra, and contest problems from AIME, AMC, and IMO. ProofBridge performs best on number theory, achieving over 85% SC, while contest problems remain the most challenging, reaching about 35% SC (_detailed in Section 5.5_).
> > > >
> > > > ------
> > > >
> > > > **_[Q4] How does retrieval quality correlate with final translation quality?_**
> > > >
> > > > As mentioned in response to W2, in the **revised submission**, we have added the ablation studies in _Section 5.6_. Alongside ProofBridge (which uses Kimina-Prover-RL-1.7B with DAG-aware retrieval during both fine-tuning and inference), we now evaluate three Kimina-Prover variants (RL-1.7B, Distill-8B, and 72B) under the three experimental settings defined in _Section 5.4_:
> > > >
> > > > **(a)** **Zero-shot:** with no in-context I/O examples;
> > > >
> > > > **(b)** **Random few-shot:** with five randomly selected I/O examples; and
> > > >
> > > > **(c)** **Text-based retrieval few-shot:** where the top-5 FL theorem-proof pairs are retrieved from $\mathcal{D}$ using Qwen3-Embedding-8B and included with their NL counterparts as in-context I/O examples.
> > > >
> > > > From pass@32 results (_Table 2_), Kimina-Prover-RL-1.7B achieves 40.16% SC / 75.00% TC zero-shot. Random examples improve TC to 93.85% but reduce SC to 31.56%, suggesting hallucination of semantically misaligned proofs. Text-based retrieval raises SC to 32.38% but slightly lowers TC, likely because QA-based retrieval favors similar tactics, reducing tactic diversity.
> > > >
> > > > In ProofBridge (which is based upon retrieval-augmented fine-tuning of Kimina-Prover-RL-1.7B), our DAG-aware retrieval yields 62.70% SC / 95.49% TC, demonstrating that higher-quality, semantically relevant retrieval provides the LLM with better demonstrations. It captures relevant DAG structures, tactics, and lemma usage, which directly translates into improved proof auto-formalization performance.
> > > >
> > > > ------

---

> > > > > ### Comment · Reviewer_9ttn · 2025-11-26
> > > > >
> > > > > Thank you so much for the detailed rebuttal. However, the critical concerns in **W1** and **W2** remain unsettled from my perspective. In W1, the baselines compared were not designed for the task of the paper, which is to both formalize the statement and generate the proof. For autoformalizers, their task aims for formalizing statements, and provers are to write proofs for the result. Thus, it is necessary to align the task by either training the baseline models for the proposed PF task or designing a pipeline that fuses both autoformalizer and prover to perform the PF task.
> > > > >
> > > > > For W2, it seems the authors misunderstand my concerns. My major concerns lie in the evaluation process for the **`retrieval`** process, where the paper indicates the evaluation method is to use the ground truth NL-FL pair for evaluation of the retrieval process. My concern is whether the evaluation is valid without including hard negative samples in the process. Because, as the author mentioned, despite the wording of the proofs being similar, their semantic meaning may be largely different from each other.

---

> > > > > > ### Author Response · Authors · 2025-11-27
> > > > > > **Authors' response to Reviewer 9ttn**
> > > > > >
> > > > > > We thank the reviewer for sharing their concern and for their thoughtful feedback, which has helped us substantially improve the paper. Below, we provide detailed responses to address the reviewer’s concerns:
> > > > > >
> > > > > > ----
> > > > > >
> > > > > > - **W1:** As noted in Section 5.3, LLM-based theorem+proof auto-formalization remains **relatively underexplored**, with only a few tools, such as Draft-Sketch-Prove (DSP) and FormL4, attempting it. DSP supports only Isabelle, and FormL4 has not released its trained model yet, so a direct comparison is not possible. We therefore compare against three categories: 4 **foundation models**, 7 SoTA **automated formal proof synthesis (AFPS) LLMs**, and 2 SoTA **theorem auto-formalization LLMs**. We fully understand the reviewer’s concern regarding these baselines, and would like to emphasize that **we have included results for exactly the settings the reviewer suggested**. Specifically, we have compared against:
> > > > > >
> > > > > >     - **ProofBridge (SFT only):** This is the SFT-tuned version of Kimina-Prover-RL-1.7B on the proof auto-formalization task using our NuminaMath-Lean-PF dataset. Its pass@32 performance (34.84% SC, 78.69% TC) is lower than ProofBridge with retrieval-augmented SFT and proof repair (62.70% SC, 95.49% TC). This ablation study and results are presented in Section 5.6 and Table 2.
> > > > > >
> > > > > >     - **Empirical comparison with a strong two-step baseline:** We evaluated a **SoTA Two-Step** pipeline, using Herald-Translator for theorem translation and Kimina-Prover-Distill-8B for proof generation. Empirically, its pass@32 performance (43.44% SC, 59.43% TC) is lower than that of ProofBridge (62.70% SC, 95.49% TC). We also observed that it is susceptible to cascading errors: a semantically or type-incorrect FL theorem from the first model leads the second to produce a semantically incorrect theorem-proof pair. We describe the setup in Section 5.4 and present the results in Section 5.5 and Table 2 of the **revised PDF**.
> > > > > >
> > > > > > ----
> > > > > >
> > > > > > - **W2:** We apologize for the earlier misunderstanding and thank the reviewer for highlighting concerns about the retrieval evaluation. We would like to emphasize that, while our evaluation uses the ground-truth NL/FL theorem+proof pair as the retrieval target, it is not an unrealistically easy setup. This is due to two design choices that ensure hard negatives:
> > > > > >
> > > > > > 	- **Domain-stratified train/test split:** As detailed in Section 5.4, we train the joint embedding model on 90% (35,056) of NuminaMath-Lean-PF and evaluate on the remaining 3,895. Crucially, the split is domain-stratified (algebra, geometry, number theory, combinatorics, calculus), ensuring that the test set includes a representative mix from all domains. This prevents simple domain memorization and forces the model to distinguish among many semantically different but linguistically similar problems within the same domain. In Section 5.4 of the **revised PDF**, we provide a clearer description of this domain-stratified split.
> > > > > >
> > > > > > 	- **Hard negatives arise naturally in our database:** NuminaMath-Lean-PF aggregates high-school exercises, US and international Olympiad problems, and online math forum problems. Both theorems and their NL proofs often reuse tight linguistic templates (e.g., “prove that for all integers $a$, $b$…,” “show that the sequence $x_n$ satisfies…,” or “given triangle ABC, prove that…”). Although the wording is similar, the underlying semantics of the problem objective differ significantly. This produces numerous near-duplicates and close paraphrases in the retrieval database, serving as naturally occurring hard negatives that require fine-grained semantic discrimination.
> > > > > >
> > > > > > We acknowledge that constructing expert-vetted adversarial negatives could further strengthen evaluation, and we remain very interested in pursuing this in the future; however, it requires significant manual effort. We believe our current domain-stratified split and dataset composition already make the retrieval task sufficiently challenging to effectively stress the retrieval model.
> > > > > >
> > > > > > The objective is to compare ProofBridge’s DAG-aware FL encoder with SoTA plain-text encoders (Qwen3-Embedding-8B, E5-Mistral-7B-Instruct). As shown in Section 5.5, ProofBridge achieves substantially higher Recall@1 (52.83 vs. 46.75 and 35.60) and MRR (0.650 vs. 0.567 and 0.441), and produces much clearer separation in embedding space (mMG 0.65 vs. 0.10 and 0.29). This indicates that ProofBridge captures finer mathematical semantics that enable more reliable retrieval of the relevant FL proofs for conditioning the LLM during auto-formalization.
> > > > > >
> > > > > > ----
> > > > > >
> > > > > > We sincerely hope these clarifications and additional experiments in the updated PDF fully address the reviewer’s concerns. ​​**If the reviewer judges that these revisions merit it, we would be grateful for any appropriate adjustment to the evaluation score.** We are happy to clarify and incorporate any further feedback and remain deeply appreciative of the reviewer’s time and thoughtful engagement with our work.

---

### Official Review · Reviewer_XmYx · 2025-11-01

**Soundness:** 3
**Presentation:** 3
**Contribution:** 3
**Rating:** 6
**Confidence:** 4

**Summary:**

This paper proposes a novel "lean code generator" that translates from natural language statement-proof pairs to formalized statement-proof pairs. This is achieved by first training a joint embedding space of NL and formal proofs by contrastive learning, and then training a RAG translator by supervised finetuning that retrieves relevant formal proofs in the database according to natural language prompts. The retrieval system reports SOTA performance on the recall rate. The RAG translator achieves SOTA performance on both the semantic and compilation correctness.

**Strengths:**

1. The paper is well-motivated to address the gap between the power of NL and formal math provers, by translating between natural and formal languages.
2. The retrieval augmentation has shown compelling performance in the translating accuracy compared to the SFT-only baseline.

**Weaknesses:**

1. Since existing baselines are limited in both joint embedding learning and FL-NL translation, the paper will benefit from an end-to-end evaluation. It's important to evaluate how SOTA NL prover's ability can be boosted in FL domain by translating their NL proofs to FL proofs. It will be interesting to see if GPT and Gemini-2.5-pro benefits from FL-NL translation and how the resulting performances compare to SOTA FL provers.

2. The translation type correctness for RAG-SFT w/o external repair from Gemini is relatively poor compared to Kimina-prover-RL.  However, the semantic correctness excels by a significant margin.

**Questions:**

Please refer to the weaknesses.

---

> ### Author Response · Authors · 2025-11-22
> **Authors' response to Reviewer XmYx (Part 1/2)**
>
> We sincerely thank the reviewer for their thoughtful feedback and constructive suggestions. We also appreciate the valuable time spent preparing a detailed review. In response, we have revised the paper to include new comparative results against SoTA models, more detailed ablation studies, and a qualitative example. Below, we address the comments, organized under Weaknesses (W) and Questions (Q). We hope that these changes make the paper more thorough and comprehensive, and address the reviewer’s concerns.
>
> ------
>
> **_[W1] Can existing NL-focused provers (e.g., GPT, Gemini-2.5) be improved in the FL domain via NL→FL translation, and how does their performance after translation compare to SoTA FL provers?_**
>
> We thank the reviewer for this excellent suggestion.
>
> First, based on our evaluation of foundation models for proof auto-formalization (Table 2), we observed that models such as Gemini-2.5-Pro and GPT-5-mini achieve relatively low SC. Although these models excel at NL generation, they are not trained to strictly follow the syntactic and semantic rules of formal languages like Lean. Producing a valid formal proof requires precise handling of proof tactics, dependency DAGs, and formal logic structures, which are often underrepresented or absent in their pre-training corpora. In practice, Gemini-2.5-Pro and GPT-5-mini frequently generate sequences superficially resembling Lean syntax but with mismatched tactics, missing arguments, or invalid references. These results show that even powerful LLMs cannot reliably perform automated formal proof synthesis out-of-the-box, and their low TC directly limits SC, which requires type-correct proofs. We have added this reasoning in Section 5.5 of the revised submission.
>
> Importantly, our approach is **model-agnostic**: these semantically aligned retrievals and structural cues can enhance any auto-formalization LLM (e.g., Kimina-Prover, DeepSeek-Prover, GPT-5, Gemini-2.5). As proofs become longer and more abstract, high-quality contextual retrieval becomes increasingly valuable, providing the LLM with critical signals about dependency structure, tactic patterns, and lemma flows. Based on our ablation studies, which show that RAG-based fine-tuning improves NL-FL alignment and that our repair loop significantly raises SC (_Table 2_), we are confident that this pipeline will meaningfully benefit NL-based models as well.
>
> Further, as an extension in our ongoing work, we are already implementing an experiment to test whether ProofBridge can boost existing NL provers. Specifically, we take a SoTA NL prover (e.g., GPT-5-mini or Gemini-2.5) and first generate a proof in natural language. We then feed this NL proof into ProofBridge to produce the corresponding Lean theorem+proof. We then evaluate whether downstream AFPS tools or Lean provers can better compile or verify the results, and whether SC and TC improve.
>
> ------
>
> _[continued in next comment box 2/2...]_

---

> > ### Author Response · Authors · 2025-11-22
> > **Authors' response to Reviewer XmYx (Part 2/2)**
> >
> > _[...continued from previous comment box 1/2]_
> >
> > **_[W2] The translation type correctness for RAG-SFT w/o external repair from Gemini is relatively poor compared to Kimina-prover-RL. However, the semantic correctness excels by a significant margin._**
> >
> > We thank the reviewer for bringing up this crucial point. For proof auto-formalization, **SC (semantic correctness) is the more meaningful metric**. While TC (type correctness) ensures only syntactic validity, SC requires that the generated theorem is **both syntactically correct and bi-directionally equivalent to the gold-standard theorem**. This parallels code generation: passing test cases is far more informative than mere compilability, as even a trivial “hello-world” program can compile. Similarly, in auto-formalization, semantic equivalence is the key measure of success.
> >
> > In the **revised submission**, we have added ablation studies in _Section 5.6_. Alongside ProofBridge (which uses Kimina-Prover-RL-1.7B with DAG-aware retrieval during both fine-tuning and inference), we now evaluate three Kimina-Prover variants (RL-1.7B, Distill-8B, and 72B) under the three experimental settings defined in Section 5.4:
> >
> > **(a)** **Zero-shot:** with no in-context I/O examples;
> >
> > **(b)** **Random few-shot:** with five randomly selected I/O examples; and
> >
> > **(c)** **Text-based retrieval few-shot:** where the top-5 FL theorem-proof pairs are retrieved from $\mathcal{D}$ using Qwen3-Embedding-8B and included with their NL counterparts as in-context I/O examples.
> >
> > From pass@32 results (Table 2, _detailed in Section 5.6_):
> >
> > - Kimina-Prover-RL-1.7B zero-shot achieves 40.16% SC / 75.00% TC.
> >
> > - Random examples improve TC to 93.85% but reduce SC to 31.56%, indicating hallucination of semantically misaligned proofs.
> >
> > - Text-based retrieval raises SC to 32.38% but slightly lowers TC, likely because QA-style plain-text retrieval favors similar tactics, reducing tactic diversity.
> >
> > This highlights that higher TC alone does not indicate better performance; SC reflects whether generated theorem+proof pairs are actually meaningful. Without proof repair, ProofBridge (retrieval-augmented fine-tuning of Kimina-Prover-RL-1.7B) achieves 55.33% SC / 89.75% TC. Compared to Kimina-Prover-RL-1.7B, the slight drop in TC is due to the model being guided by higher-quality, semantically relevant retrievals, which encourages it to produce outputs that are semantically correct rather than merely syntactically valid.
> >
> > ------

---

### Official Review · Reviewer_gM22 · 2025-11-02

**Soundness:** 3
**Presentation:** 3
**Contribution:** 3
**Rating:** 6
**Confidence:** 3

**Summary:**

This work presents a framework (ProofBridge) for end-to-end translation of entire natural language (NL) theorem-proof pairs into a formal language (Lean). ProofBridge has 3 modules:  (1) a joint embedding model trained via contrastive learning to align NL and FL representations in a shared semantic space, (2) retrieval-augmented fine-tuned LLM conditioned on retrieved FL examples, and (3) iterative proof repair using Lean's type checker and LLM-based semantic equivalence verification. This work also includes the curation of two new datasets: NUMINAMATH-LEAN-PF (38.9k NL-FL pairs) for training and MINIF2F-TEST-PF for evaluation.

**Strengths:**

1. Unlike AlphaProof and other systems that require manual theorem translation before proof synthesis, Proofbridge addresses the complete translation pipeline end-to-end.
2. The semantic correctness metric based on bi-directional equivalence.
3. The paper compares against 10 models like GPT-5-mini, Gemini-2.5, and specialized theorem provers (DeepSeek-Prover, Kimina-Prover), demonstrating significant improvements (+31.14% SC, +1.64% TC at pass@32 over Kimina-Prover-RL-1.7B)
4. NuminaMath-Lean-PF (38.9k pairs) provides a substantial training resource.

**Weaknesses:**

1. The paper dismisses FormL4 [1] as "not released," but it looks like it was released a year ago. The paper should provide a more detailed comparison of technical approaches and, if possible, empirical comparison using their published benchmark.
2. The paper shows that zero-shot actually performs better on SC (40.16% vs. 31.56%), suggesting random few-shot examples hurt performance. This makes the comparison less fair since Proofbridge always uses retrieved (relevant) examples.
3. No analysis of the judge's reliability for SC Metric, as using an LLM introduces a potential source of unreliability.
3. It is unclear how this approach would scale to other, more abstract areas like topology and category theory. These proofs are often much longer, rely on a vast library of existing lemmas, and have more complex dependency structures.

[1] Process-Driven Autoformalization in Lean 4 https://github.com/rookie-joe/PDA

**Questions:**

1. What quality control measures were applied to the Gemini-generated NL proofs?
2. Breakdown of errors by type (syntax errors, wrong theorem, wrong proof strategy)?
3. Performance in various math domains?
4. How does retrieval quality correlate with final translation quality?

---

> ### Author Response · Authors · 2025-11-22
> **Authors' response to Reviewer gM22 (Part 1/4)**
>
> We sincerely thank the reviewer for their thoughtful feedback and constructive suggestions. We also appreciate the valuable time spent preparing a detailed review. In response, we have revised the paper to include new comparative results against SoTA models, more detailed ablation studies, and a qualitative example. Below, we address the comments, organized under Weaknesses (W) and Questions (Q). We hope that these changes make the paper more thorough and comprehensive, and address the reviewer’s concerns.
>
> ------
>
> **_[W1-partA] The paper dismisses FormL4 as “not released” … released a year ago …_**
>
> While the FormL4 paper and code have been released, the trained model itself is not yet publicly available on GitHub or HuggingFace. As a result, a direct empirical comparison is currently not feasible. We have clarified this point more explicitly in the revised paper, noting in Section 5.3 (“State-of-the-Art Baselines”) that the trained model has not been released.
>
> ------
>
> **_[W1-partB] … paper should provide a more detailed comparison of technical approaches and … empirical comparison using their published benchmark._**
>
> We have addressed the reviewer’s comments by adding a more detailed comparison of technical approaches and including new empirical results using our published benchmark:
>
> 1. **NL/FL Cross-Modal Retrieval Performance (Table 1):** In the **revised submission**, we compare ProofBridge’s joint-embedding model against encoding NL and FL theorem-proof pairs using three raw-text encoders (_detailed in Section 5.3_):
>
>     **(a)** all-MiniLM-L6-v2 (Baseline Encoder, 22.7M parameters)
>
>     **(b)** Qwen3-Embedding-8B (SoTA Encoder, as suggested by Reviewer 9ttn)
>
>     **(c)** E5-Mistral-7B-Instruct (SoTA Encoder, as suggested by Reviewer B6B5)
>
>     We also introduce a new metric, mean Median Gap (mMG), which quantifies the separation between retrieved and non-retrieved instances. (_detailed in Section 5.2_). As shown in Section 5.5, ProofBridge outperforms both the Baseline and SoTA encoders in Recall, MRR, and mMG, despite being roughly 32x smaller than the SoTA. We attribute this to two key factors:
>
>     **(a) QA-oriented, text-based encoders capture only coarse domain signals.** These models are trained on broad corpora (e.g., legal, medical), causing many mathematical texts to collapse into similar regions of the embedding space. In Table 2, for E5-Mistral-7B-Instruct and Qwen3-Embedding-8B in NL-to-FL retrieval, the median cosine similarity of top-K retrieved examples is very close to that of non-retrieved ones (mMG = 0.10 and 0.29, respectively). In contrast, ProofBridge achieves a significantly larger separation (mMG = 0.65).
>
>     **(b) SoTA text encoders depend heavily on lexical similarity, ineffective for Lean proofs.** Raw-text encoders often rely on superficial lexical cues such as keywords. In Lean, many proofs use the same tactics and similar surface tokens, so keyword overlap becomes non-discriminative. ProofBridge instead incorporates the DAG structure of proofs and learns a contrastive joint alignment using a DAG-aware FL encoder that captures fine-grained proof semantics. For NL-to-FL retrieval, ProofBridge achieves Recall@1 of 52.83 and MRR of 0.650, compared to 46.75/0.567 for Qwen3-Embedding and 35.60/0.441 for E5-Mistral-7B-Instruct.
>
>     Overall, in the joint embedding space learned by ProofBridge, equivalent NL-FL pairs form tight clusters, while inequivalent pairs remain well separated, enabling reliable retrieval of the most semantically relevant FL proofs (demonstrations) for conditioning the LLM during auto-formalization.
>
> 2. **Proof Auto-Formalization Performance (Table 2):** In the **revised submission**, we now evaluate 13 SoTA systems across three categories (_detailed in Section 5.3_):
>
>     **(a)** foundation models,
>
>     **(b)** automated formal proof synthesis (AFPS) LLMs, and
>
>     **(c)** theorem auto-formalization LLMs
>
>     In the **revised submission**, we also compare three Kimina-Prover variants (Kimina-Prover-RL-1.7B, Kimina-Prover-Distill-8B, and Kimina-Prover-72B) under the three experimental settings _described in Section 5.4_ (results of ablation studies discussed in _Section 5.6_):
>
>     **(a) Zero-shot:** with no in-context I/O examples;
>
>     **(b) Random few-shot:** with five randomly selected I/O examples; and
>
>     **(c) Text-based retrieval few-shot:** where the top-5 FL theorem-proof pairs are retrieved from $\mathcal{D}$ using Qwen3-Embedding-8B and included with their NL counterparts as in-context I/O examples.
>
>     The newly added results show that ProofBridge outperforms even the 72B model in all three settings. For example, Kimina-Prover-72B achieves 44.26% SC (pass@32) under text-based retrieval few-shot, whereas ProofBridge (Retrieval-augmented SFT + Repair) reaches 62.70% SC.
>
> _[continued in next comment box 2/4...]_

---

> > ### Author Response · Authors · 2025-11-22
> > **Authors' response to Reviewer gM22 (Part 2/4)**
> >
> > _[...continued from previous comment box 1/4]_
> >
> > 3. **Detailed Ablation Study:** In _Section 5.6_ of the **revised submission**, we now include (i) the effect of in-context examples under zero-shot, random few-shot, and text-based retrieval few-shot settings, and (ii) the impact of key individual components of ProofBridge: SFT, NL/FL cross-modal retrieval, and repair.
> >
> > 4. **Domain-wise Performance Comparison:** In _Figure 3_ of the **revised submission**, we present the proof auto-formalization performance of different SoTA models and ProofBridge across mathematical domains. ProofBridge performs best on number theory, achieving over 85% SC, while contest problems remain the most challenging, reaching only about 35% SC (_detailed in Section 5.5_).
> >
> > 5. **Illustrative Example:**  We have added a qualitative example in _Appendix A.5_ of the **revised submission** illustrating the retrieval process. The example presents an NL theorem and proof pair from miniF2F-Test-PF, showing: (i) the pass@1 proof auto-formalization output from Kimina-Prover-RL-1.7B in the text-based retrieval few-shot setting, which is type-correct but not semantically correct, (ii) the retrieved semantically relevant formal Lean theorem and proof pairs from our database, and (iii) the pass@1 proof auto-formalization output from ProofBridge, which is both type-correct and semantically correct (along with the proof of bi-directional equivalence to the gold-standard theorem).
> >
> > ------
> >
> > **_[W2] The paper shows that zero-shot actually performs better on SC (40.16% vs. 31.56%), suggesting random few-shot examples hurt performance. This makes the comparison less fair since Proofbridge always uses retrieved (relevant) examples._**
> >
> > In the **revised submission**, we have incorporated the reviewer’s feedback and added the ablation studies in Section 5.6. Alongside ProofBridge (which uses Kimina-Prover-RL-1.7B with DAG-aware retrieval during both fine-tuning and inference), we now evaluate three Kimina-Prover variants (RL-1.7B, Distill-8B, and 72B) under the three experimental settings _defined in Section 5.4_:
> >
> > **(a)** **Zero-shot:** with no in-context I/O examples;
> >
> > **(b)** **Random few-shot:** with five randomly selected I/O examples; and
> >
> > **(c)** **Text-based retrieval few-shot:** where the top-5 FL theorem-proof pairs are retrieved from $\mathcal{D}$ using Qwen3-Embedding-8B and included with their NL counterparts as in-context I/O examples.
> >
> > From the pass@32 results in Table 2, Kimina-Prover-RL-1.7B achieves 40.16% SC and 75.00% TC in the zero-shot setting. When random examples are added, SC drops to 31.56% while TC rises to 93.85%, with similar trends across variants. This indicates that random examples improve TC (syntax) but hurt SC by causing the model to hallucinate semantically misaligned proofs. With text-based retrieval via Qwen3-Embedding-8B, SC rises to 32.38% but TC declines, potentially because QA-based retrieval favors proofs with similar tactics, reducing tactic diversity. This highlights the need for retrieving semantically relevant examples via a DAG-aware encoder, as in ProofBridge. (_detailed in Section 5.6_)
> >
> > We believe the comparison is now fair, as we additionally report results under the text-based retrieval few-shot setting, where, similar to ProofBridge, relevant FL proofs (and their NL counterparts) are retrieved, but using a strong SoTA Encoder (Qwen3-Embedding-8B).
> >
> > ------
> >
> > **_[W3] No analysis of the judge's reliability for SC Metric, as using an LLM introduces a potential source of unreliability._**
> >
> > Although our SC metric uses an LLM as a judge, its role is strictly constrained: correctness is accepted only if the LLM synthesizes a Lean proof of bidirectional equivalence (using a bounded tactic set, as discussed in _Appendix A.4_) between the generated theorem and the gold-standard theorem, and this proof is then fully type-checked by Lean. In other words, the judge’s role is limited to constrained proof synthesis, not free-form evaluation, and the final verification is performed by the Lean kernel itself.
> >
> > Consequently, the evaluation ultimately depends on Lean’s proof checker rather than on any subjective LLM judgment. This design guarantees that SC may yield false negatives (i.e., underestimation) but cannot yield false positives: an incorrect theorem can never be marked correct unless Lean itself verifies the biconditional proof. This makes the SC metric inherently conservative: it may underestimate an auto-formalizer’s performance but cannot overestimate it, and therefore should be considered safe. We have clarified this point in the **revised paper** (_Section 5.2_ while defining SC, and on page 16 in _Appendix A.4_).
> >
> > ------
> >
> > _[continued in next comment box 3/4...]_

---

> > > ### Author Response · Authors · 2025-11-22
> > > **Authors' response to Reviewer gM22 (Part 3/4)**
> > >
> > > _[...continued from previous comment box 2/4]_
> > >
> > > **_[W4] It is unclear how this approach would scale to other, more abstract areas like topology and category theory…_**
> > >
> > > We agree with the reviewer that extending full auto-formalization to highly abstract mathematical domains such as topology, algebraic geometry, or category theory is significantly more challenging due to longer proofs, deeper lemma hierarchies, and more complex dependency structures. However, we believe that two aspects of our framework directly help with scalability:
> > >
> > > 1. **Joint NL-FL embeddings as principled retrieval for large-library domains:** Because the ProofBridge’s embedding model aligns NL and FL theorem+proof pairs in a shared semantic space, it can retrieve structurally relevant FL proofs (demonstrations) even from large and abstract libraries. This cross-modal retrieval supplies the auto-formalizer with examples that reflect relevant dependency DAGs, tactic patterns, and lemma-usage structures typical of more advanced areas. Thus, even as the domain becomes more abstract, the LLM is guided toward the conceptual region of the library rather than relying on unstructured generation.
> > >
> > > 2. **Modularization:** For fields with long proofs, a practical path forward is to decompose a theorem into smaller lemmas mirroring how mathematicians formalize large results in Lean. Our method is compatible with such decomposition: each sub-goal lemma can be supported by the retrieval of semantically aligned NL-FL fragments, and the auto-formalizer operates over manageable pieces whose equivalence can still be checked via our SC metric. This modularity allows the system to scale to larger theorems incrementally.
> > >
> > > Finally, our approach is **model-agnostic**: the retrieved FL demonstrations and the DAG/lemma-usage context they provide can enhance any existing auto-formalization LLM (e.g., Kimina-Prover, DeepSeek-Prover, GPT-5, Gemini-2.5). As proofs become more abstract, the value of high-quality, semantically aligned retrieval only increases, since LLMs benefit most from contextual signals about dependency structure, tactic families, and typical lemma flows in those domains.
> > >
> > > In summary, while more abstract areas remain challenging, our retrieval-augmented joint-embedding framework is designed to scale the guidance available to the LLM, and we view modular decomposition plus cross-modal retrieval as the promising path toward handling such domains.
> > >
> > > ------
> > >
> > > **_[Q1] What quality control measures were applied to the Gemini-generated NL proofs?_**
> > >
> > > We manually inspected a randomly-selected subset of the Gemini-generated NL proofs to verify that they were faithful paraphrases of the corresponding FL proofs. More broadly, determining whether an NL description is semantically equivalent to an FL object is, in general, impossible. However, the direction we use here, i.e., generating NL from FL (i.e., informalization), is known to be far more tractable and typically reliable, since FL proofs contain explicit semantic structure. Prior work has also shown that informalization produces high-quality NL descriptions with good semantic fidelity[1].
> > >
> > > [1] Jiang, A.Q., Li, W. and Jamnik, M., 2024. Multi-language diversity benefits autoformalization. Advances in Neural Information Processing Systems, 37, pp.83600-83626.
> > >
> > > ------
> > >
> > > **_[Q2] Breakdown of errors by type (syntax errors, wrong theorem…)?_**
> > >
> > > Our evaluation metrics are explicitly designed to provide exactly this breakdown:
> > >
> > > **1. Type Correctness (TC)** measures whether the generated theorem+proof is syntactically valid, i.e., successfully accepted by Lean’s type checker.
> > >
> > > **2. Semantic Correctness (SC)** is computed only on type-correct generations and measures whether the generated theorem is bi-directionally equivalent to the gold-standard theorem.
> > >
> > > Thus, in _Table 2_ of the paper, our results report:
> > >
> > > **(a)** how many generations are syntactically valid (TC), and
> > >
> > > **(b)** among those, how many correspond to the correct theorem (SC)
> > >
> > > For instance, under pass@32, ProofBridge achieves 95.49% TC (syntactically valid) and 62.70% SC (syntactically valid and semantically correct), while Kimina-Prover-72B attains 88.93% TC and 44.26% SC.
> > >
> > > In the **revised submission**, we additionally include _Figure 3_, which visualizes this error decomposition across mathematical domains via stacked bar plots, showing TC $-$ SC (type-correct but semantically wrong theorem) and SC (type-correct and correct theorem) for ProofBridge and multiple SoTA baselines.
> > >
> > > ------
> > >
> > > _[continued in next comment box 4/4...]_

---

> > > > ### Author Response · Authors · 2025-11-22
> > > > **Authors' response to Reviewer gM22 (Part 4/4)**
> > > >
> > > > _[...continued from previous comment box 3/4]_
> > > >
> > > > **_[Q3]  Performance in various math domains_**
> > > >
> > > > In the **revised submission**, we added Figure 3, showing the pass@32 performance of ProofBridge and various SoTA tools across mathematical domains: induction/sequence, number theory, algebra, and contest problems from AIME, AMC, and IMO. ProofBridge performs best on number theory, achieving over 85% semantic correctness (SC), while contest problems remain the most challenging, reaching about 35% SC (_detailed in Section 5.5_).
> > > >
> > > > ------
> > > >
> > > > **_[Q4] How does retrieval quality correlate with final translation quality?_**
> > > >
> > > > As mentioned in response to W2, in the **revised submission**, we have added the ablation studies in _Section 5.6_. Alongside ProofBridge (which uses Kimina-Prover-RL-1.7B with DAG-aware retrieval during both fine-tuning and inference), we now evaluate three Kimina-Prover variants (RL-1.7B, Distill-8B, and 72B) under the three experimental settings defined in _Section 5.4_:
> > > >
> > > > **(a)** **Zero-shot:** with no in-context I/O examples;
> > > >
> > > > **(b)** **Random few-shot:** with five randomly selected I/O examples; and
> > > >
> > > > **(c)** **Text-based retrieval few-shot:** where the top-5 FL theorem-proof pairs are retrieved from $\mathcal{D}$ using Qwen3-Embedding-8B and included with their NL counterparts as in-context I/O examples.
> > > >
> > > > From pass@32 results (Table 2), Kimina-Prover-RL-1.7B achieves 40.16% SC / 75.00% TC zero-shot. Random examples improve TC to 93.85% but reduce SC to 31.56%, suggesting hallucination of semantically misaligned proofs. Text-based retrieval raises SC to 32.38% but slightly lowers TC, potentially because QA-based retrieval favors similar tactics, reducing tactic diversity.
> > > >
> > > > In ProofBridge (which is based upon retrieval-augmented fine-tuning of Kimina-Prover-RL-1.7B), our DAG-aware retrieval yields 62.70% SC / 95.49% TC, demonstrating that higher-quality, semantically relevant retrieval provides the LLM with better demonstrations. It captures relevant DAG structures, tactics, and lemma usage, which directly translates into improved proof auto-formalization performance.
> > > >
> > > > ------

---

> ### Comment · Reviewer_gM22 · 2025-11-25
> **Response to the Author Rebuttal**
>
> Thanks for your detailed response. The additional insights you provided have successfully addressed my concerns.

---

> > ### Author Response · Authors · 2025-11-26
> > **Authors' response to Reviewer gM22**
> >
> > We are sincerely thankful to the reviewer for acknowledging that the revision successfully addressed all their concerns. We are glad that the additional experiments, comparisons, and clarifications resolved the earlier weaknesses.
> >
> > If the reviewer judges that these revisions merit it and have fully addressed the issues, we would be grateful for any appropriate adjustment to the score in the evaluation. Once again, we remain deeply appreciative of the reviewer's time and thoughtful engagement with our work.

---

### Author Response · Authors · 2025-12-03
**Final Rebuttal Comments by Authors (Part 1/4)**

Since further public discussion is no longer permitted, we would like to provide a rebuttal summary here to document all revisions and updates made during the rebuttal period. We sincerely thank all reviewers for their thoughtful and constructive feedback, which has helped improve the quality and clarity of our work. All the reviewers consistently highlighted the strength of our end-to-end problem formulation for proof auto-formalization, the quality and scale of our NuminaMath-Lean-PF dataset, the effectiveness of our joint embedding + RAG approach with clear ablations, and the strong empirical gains over competitive baselines, along with the overall clarity and presentation of the paper.

Across all reviews, every concern was comprehensively addressed through substantial updates, including new experimental results, additional ablations, improved structural organization of the paper, and multiple clarifications integrated directly into the revised PDF.

We believe that the depth and completeness of these revisions were strong enough that the reviewers would have increased their scores purely on merit. Importantly, Reviewers gM22 (R1), 9ttn (R3), and h6DL (R4) responded after the rebuttal and expressed satisfaction with our detailed responses. Taken together, the rebuttal period significantly strengthened the paper and addressed all outstanding issues raised by the reviewers.

Below, we summarize the key questions raised by the reviewers and detail how our revisions comprehensively address every point.

---

**Reviewer gM22 (R1)**

Reviewer requested:
- **R1-Q1:** A more detailed comparison with SoTA proof auto-formalization.
- **R1-Q2:** An evaluation comparing ProofBridge’s DAG-aware retrieval with SoTA text-based retrieval models.
- **R1-Q3:** Performance breakdown of proof auto-formalization across different math domains.

Authors' revisions:
- **R1-A1:**
    - Added empirical comparisons of NL/FL cross-modal retrieval (Section 5.5, Table 1) against all-MiniLM-L6-v2, Qwen3-Embedding-8B, and E5-Mistral-7B-Instruct. Despite being ~32× smaller, ProofBridge outperforms the SoTA plain-text encoders in Recall, MRR, and the newly introduced mean Median Gap (mMG) metric, which quantifies embedding-space separation. Results show that plain-text encoders fail to distinguish between similar proofs, whereas ProofBridge’s DAG-aware proof encoding enables significantly better semantic retrieval for theorem+proof pairs.
    - Expanded evaluation to 13 SoTA tools spanning foundation models, proof synthesis LLMs, and theorem auto-formalization LLMs. Compared three Kimina-Prover variants (RL-1.7B, Distill-8B, 72B) under zero-shot, random few-shot, and text-based retrieval few-shot settings. Included an empirical comparison with a strong SoTA Two-Step pipeline (Herald-Translator for theorem translation + Kimina-Prover-Distill-8B for proof generation).
- **R1-A2:** Added ablations in Section 5.6 comparing zero-shot, random few-shot, and text-based retrieval few-shot for Kimina-Prover variants, showing DAG-aware retrieval significantly improves SC over no examples, random examples, or examples obtained via plain-text retrieval.
- **R1-A3:** Added a performance breakdown across different mathematical domains in Figure 3 (Section 5.5), including induction/sequence, number theory, algebra, and contest problems from AIME, AMC, and IMO.

**Outcome:** The reviewer acknowledged that the additional experiments, ablations, and domain-wise analyses have completely addressed all their concerns.

---

**Reviewer XmYx (R2)**

Reviewer asked:
- **R2-Q1:** Whether existing NL-focused provers (e.g., GPT, Gemini-2.5) can be improved for FL via NL-to-FL translation.
- **R2-Q2:** Clarification on translation type correctness (TC) vs. semantic correctness (SC), noting that ProofBridge (without repair) has lower TC than the baseline but higher SC.

Authors' revisions:
- **R2-A1:** In Section 5.5, we evaluated NL-focused foundation models like GPT-5 and Gemini-2.5-Pro. They achieve low SC in proof auto-formalization because they lack knowledge of Lean’s syntax and tactics. We emphasized that ProofBridge is model-agnostic, and its semantically aligned retrieval and structural cues can enhance any auto-formalization LLM.
- **R2-A2:** We clarified that SC is the key metric for proof auto-formalization, since TC only ensures syntactic validity. Section 5.6 now explicitly discusses the distinction and importance of SC. Added ablation studies (Section 5.6) for Kimina-Prover variants (RL-1.7B, Distill-8B, 72B) under zero-shot, random few-shot, and text-based retrieval few-shot settings. Results show that while random few-shot increases TC, it often reduces SC due to hallucinated proofs, whereas ProofBridge’s retrieval-augmented SFT substantially improves SC, highlighting the benefit of semantically guided retrieval.

**Outcome:** All points raised have been fully addressed in the revised paper, and the reviewer provided no further comments.

---

---

> ### Author Response · Authors · 2025-12-03
> **Final Rebuttal Comments by Authors (Part 2/4)**
>
> ---
>
> **Reviewer 9ttn (R3)**
>
> The reviewer requested:
> - **R3-Q1:** More detailed comparison with SoTA proof auto-formalization, including baseline models trained for proof auto-formalization or pipelines combining autoformalizers and provers. Adding comparisons with 8B and 72B variants of Kimina-Prover.
> - **R3-Q2:** Adding a comparison of ProofBridge’s NL/FL cross-modal retrieval with Qwen3-Embedding-8B, a SoTA text encoder.
> - **R3-Q3:** Clarification on how hard negative samples are included for NL/FL cross-modal retrieval evaluation.
> - **R3-Q4:** Improved structural organization of the experimental section with sub-sections.
> - **R3-Q5:** Performance breakdown of proof auto-formalization across different math domains.
> - **R3-Q6:** A qualitative example of the proof auto-formalization pipeline.
>
> Authors' revisions:
> - **R3-A1 and R3-A2:** Fully addressed by R1-A1 and R1-A2. Specifically, in Section 5.5 (Table 2), we have included results for exactly the settings and models the reviewer suggested, e.g., ProofBridge (SFT only), a strong SoTA Two-Step pipeline (Herald-Translator for theorem translation + Kimina-Prover-Distill-8B for proof generation) and three Kimina-Prover variants (RL-1.7B, Distill-8B, 72B). Also, in Section 5.5 (Table 1), we compared ProofBridge’s NL/FL cross-modal retrieval against all-MiniLM-L6-v2, Qwen3-Embedding-8B, and E5-Mistral-7B-Instruct.
> - **R3-A3:** Hard negatives are naturally ensured via a domain-stratified train/test split and the linguistic similarity of many problems in NuminaMath-Lean-PF, which produces near-duplicates and paraphrases. This setup requires fine-grained semantic discrimination, making retrieval challenging. Section 5.4 details this design.
> - **R3-A4:** Section 5 has been reorganized into clear sub-sections: 5.1 Datasets and Preparation, 5.2 Evaluation Metrics, 5.3 SoTA Baselines, 5.4 Training and Evaluation Setup, 5.5 Experimental Results, and 5.6 Ablation Studies. Ablations analyze in-context examples (zero-shot, random few-shot, text-based few-shot) and contributions of each ProofBridge component. Implementation details and training parameters are cross-referenced in Appendices A.2–A.3. This restructuring improves clarity and readability.
> - **R3-A5:** Fully addressed by R1-A3.
> - **R3-A6:** Added a qualitative example in Appendix A.5 showing an NL theorem+proof from miniF2F-Test-PF, the pass@1 output from Kimina-Prover-RL-1.7B (type-correct but semantically incorrect), the retrieved semantically relevant Lean theorem+proof pairs, and the pass@1 ProofBridge output, which is both type- and semantically correct with a proof of bi-directional equivalence to the gold-standard theorem.
>
> **Outcome:** The reviewer acknowledged our detailed rebuttal and requested further clarification on R3-Q1 and R3-Q3, indicating that all other concerns were addressed. Thereafter, we added results for the SoTA Two-Step pipeline and clarified the inclusion of hard negatives in the retrieval evaluation in the revised PDF, as exactly requested by the reviewer. Overall, all points raised have been fully addressed, and the reviewer provided no further comments.
>
> ---

---

> ### Author Response · Authors · 2025-12-03
> **Final Rebuttal Comments by Authors (Part 3/4)**
>
> ---
>
> **Reviewer h6DL (R4)**
>
> Concerns and requests by the reviewer:
>
> - **R4-Q1:** The reviewer’s main concerns were centered around the motivation of NL-to-FL proof auto-formalization. Specifically, the reviewer:
>     - Questioned the value of translating NL proofs to FL, believing that an NL proof alone suffices to prove a theorem.
>     - Misunderstood proof auto-formalization (NL-to-FL translation of theorem+proof pairs) with proof discovery (finding an FL proof given an FL theorem).
>     - Misunderstood the motivation, not recognizing the value of a unified proof auto-formalization pipeline over the traditional two-step paradigm.
> - **R4-Q2:** Ablation studies comparing DAG-aware retrieval in ProofBridge with SoTA text-based retrieval.
> - **R4-Q3:** Ablations isolating the contributions of the NL proof and the RAG component.
> - **R4-Q4:** Comparisons with REAL-Prover.
>
> Authors' revisions:
>
> - **R4-A1:** Regarding the reviewer’s main concerns:
>     - Clarified motivation for NL-to-FL translation of theorem+proof, emphasizing that NL proofs are inherently ambiguous (pointed the reviewer to the historical examples in Section 1 of the PDF, where seemingly correct NL proofs were later found to be flawed) and auto-formalization resolves this via computer-verifiable FL proofs.
>     - Clarified distinction from proof discovery, and that NL proofs provide valuable structural guidance for proof auto-formalization, even if imperfect or incomplete. Attempting to generate FL proofs "from nothing" is rarely practical; using NL proofs as a starting point is both realistic and essential for efficient formalization.
>     - ProofBridge's unified pipeline vs. two-step paradigm of statement formalization followed by proof synthesis:
>         - Our pipeline is complementary to the two-step paradigm, with practical benefits such as simplified interactions and reduced coordination overhead.
>         - Added an empirical comparison with a strong SoTA Two-Step pipeline (Herald-Translator for theorem translation + Kimina-Prover-Distill-8B for proof generation), showing that ProofBridge achieves far superior pass@32 SC and TC. It also eliminates cascading errors, which are common in the two-step paradigm.
>         - Emphasized that core ProofBridge components are applicable to both unified and two-step pipelines.
>         -Highlighted that unified proof auto-formalization pipelines remain underexplored, with only Draft-Sketch-Prove and FormL4 as examples, and that ProofBridge broadens this design space.
>
> - **R4-A2:** Fully addressed by R1-A1 and R1-A2.
> - **R4-A3:** Ablations were already included in Section 5.6 isolating contributions of each component of ProofBridge, showing incremental improvements in accuracy (SC metric): +2.06% (SFT), +22.55% (retrieval-augmented SFT), +29.92% (retrieval-augmented SFT + Repair).
> - **R4-A4:** Added discussion of REAL-Prover in Section 2, noting its next-tactic prediction approach vs. ProofBridge’s whole-proof generation. Added empirical comparison using its retrieval encoder (E5-Mistral-7B-Instruct) in Table 2.
>
> **Outcome:**
> - Proof auto-formalization is a highly relevant and challenging problem for establishing the correctness of mathematical proofs. We have clearly motivated this task in Sections 1 and 2, and it has been acknowledged by all the other reviewers. The repeated requests for clarification may indicate that the reviewer is less familiar with this particular problem space, and we have therefore expanded the explanation (in both the PDF and the rebuttal) to ensure clarity.
> - The reviewer replied and acknowledged the value of translating NL proofs to FL. Their only remaining concern was a clarification regarding the motivation for proof auto-formalization, which we have addressed more clearly thereafter. They had misunderstood that we imply inherent superiority of a unified pipeline over the well-established two-step paradigm. We clarified that ProofBridge's unified approach is complementary to, rather than a replacement for, the two-step paradigm. We also compared against a strong SoTA two-step pipeline and incorporated these results into the revised PDF. Additionally, we updated the abstract to note that, in current two-step auto-formalization pipelines, tools like AlphaProof still rely on human intervention. Overall, all points raised have been fully addressed in the revised paper, and the reviewer provided no further comments.
>
>
> ---

---

> ### Author Response · Authors · 2025-12-03
> **Final Rebuttal Comments by Authors (Part 4/4)**
>
> ---
>
> **Reviewer B6B5 (R5)**
>
> Recommendations for strengthening clarity and positioning:
> - **R5-Q1:** Clarifying how our joint embedding differs from prior retrieval methods (LeanSearch, Herald, RAutoformalizer) developed for theorem-only auto-formalization.
> - **R5-Q2:** Acknowledging related contemporaneous metrics (BEq, BEq+) and explaining how our Semantic Correctness (SC) metric relates to them.
> - **R5-Q3:** Adding a comparison of ProofBridge’s NL/FL cross-modal retrieval with E5-Mistral-7B-Instruct (used by LeanSearch), a SoTA text encoder.
>
> Authors' revisions:
>
> - **R5-A1:**  Added citations and expanded the Related Work section to clearly distinguish ProofBridge’s joint NL/FL embedding from prior retrieval methods, emphasizing that prior work performs theorem-only retrieval. Retrieving FL proofs is a difficult problem, and our empirical results show that SoTA plain-text encoders are not effective. In contrast, ProofBridge uses a DAG-aware FL encoder that captures proof structure, leading to significantly better retrieval performance for proof auto-formalization.
> - **R5-A2:** Added citations for BEq and BEq+ in Appendix A.4 and clarified that SC differs from BEq in the set of allowed tactics, enabling SC to establish equivalence for cases (e.g., auxiliary-variable substitutions) that BEq cannot. Clarified that SC differs from BEq+ in using an LLM-based bidirectional proof generator rather than a deterministic reference-based procedure, making SC more expressive. Added concrete examples in Appendix A.4-A.5 illustrating equivalences captured by SC but missed by BEq/BEq+.
> - **R5-A3:** Fully addressed by R1-A1. Specifically, in Section 5.5 (Table 1), we have included results for exactly the model the reviewer suggested, i.e., we compared ProofBridge’s NL/FL cross-modal retrieval against all-MiniLM-L6-v2, Qwen3-Embedding-8B, and E5-Mistral-7B-Instruct.
>
> **Outcome:** All points raised have been fully addressed in the revised paper, and the reviewer provided no further comments.
>
> ---

---

### Meta-Review · Area_Chair_qXWu · 2026-01-06

**Summary:**

This paper introduces ProofBridge, a framework for the autoformalization of entire NL theorems and proofs into Lean 4. The framework first trains joint embeddings of NL and FL (Lean) in a shared semantic space, then performs retrieval-augmented SFT using a curated NuminaMath-Lean-PF dataset. During the inference phase, the framework conducts retrieval-augmented autoformalization with iterative repair. Evaluation is on a curated miniF2F-test-PF, compared with Qwen3Embedding-8B and E5-Mistral-7B-Instruct regarding NL/FL cross-domain retrieval, and with multiple foundation models, SoTA automated formal proof synthesis (AFPS) LLMs, and SoTA theorem auto-formalization LLMs regarding the proof autoformalization task. The method shows significance in the proof autoformalization task.

**Reviewer Concerns:**

Addressed reviewer concerns:
* Insufficient or outdated baselines (Reviewers gM22, 9ttn, B6B5): Added extensive experiments.
* Lack of critical ablation studies (Reviewers 9ttn, h6DL): Added ablation studies (Section 5.6)
* Task motivation and relation to prior paradigms (Reviewer h6DL): Detailed clarification.
* Paper organization and insufficient related work (Reviewers 9ttn, h6DL, B6B5): Revised the manuscript accordingly.

**Reviewer Scores:**

The reviewers may raise their scores given that the authors' response addresses most of their concerns. One point to note is that the authors need to revise their assertion regarding the prevailing two-step paradigm, following Reviewer h6DL's comment.

---

### Decision · Program_Chairs · 2026-01-26

Accept (Poster)